**\*For correspondence:**
go.quan.8w@kyoto-u.ac.jp (QW);
matsuzaki.fumio.2w@Kyoto-u.ac.jp (FM)

**Present address:** [†]The Institute for Quantitative Biosciences, The University of Tokyo, Tokyo, Japan; [‡]Department of Aging Science and Medicine, Medical Innovation Center, Graduate School of Medicine, Kyoto University, Kyoto, Japan; [§]Department of Aging Science and Medicine, Medical Innovation Center, Graduate School of Medicine, Kyoto University, Kyoto, Japan; [#]Department of Human Morphology, Graduate School of Medicine, Dentistry and Pharmaceutical Sciences, Okayama University, Okayama, Japan; [¶]Division of Molecular and Medical Genetics, Center for Gene and Cell Therapy, The Institute for Medical Science, The University of Tokyo, Tokyo, Japan; [\*\*]Molecular Life History Laboratory, Department of Genomics and Evolutionary Biology, National Institute of Genetics, Mishima, Shizuoka, Japan; [††]Department of Aging Science and Medicine, Medical Innovation Center, Graduate School of Medicine, Kyoto University, Kyoto, Japan

# Truncated radial glia as a common precursor in the late corticogenesis of gyrencephalic mammals

**Merve Bilgic**[1,2†], **Quan Wu**[1\*‡], **Taeko Suetsugu**[1§], **Atsunori Shitamukai**[1#], **Yuji Tsunekawa**[1¶], **Tomomi Shimogori**[3], **Mitsutaka Kadota**[4], **Osamu Nishimura**[4], **Shigehiro Kuraku**[4\*\*], **Hiroshi Kiyonari**[5], **Fumio Matsuzaki**[1,2\*††]

[1]Laboratory for Cell Asymmetry, RIKEN Center for Biosystems Dynamics Research, Kobe, Japan; [2]Laboratory of Molecular Cell Biology and Development, Department of Animal Development and Physiology, Graduate School for Biostudies, Kyoto University, Kyoto, Japan; [3]Molecular Mechanisms of Brain Development, RIKEN Center for Brain Science, Wako, Japan; [4]Laboratory for Phyloinformatics, RIKEN Center for Biosystems Dynamics Research, Kobe, Japan; [5]Laboratory for Animal Resources and Genetic Engineering, RIKEN Center for Biosystems Dynamics Research, Kobe, Japan

**Abstract** The diversity of neural stem cells is a hallmark of the cerebral cortex development in gyrencephalic mammals, such as Primates and Carnivora. Among them, ferrets are a good model for mechanistic studies. However, information on their neural progenitor cells (NPC), termed radial glia (RG), is limited. Here, we surveyed the temporal series of single-cell transcriptomes of progenitors regarding ferret corticogenesis and found a conserved diversity and temporal trajectory between human and ferret NPC, despite the large timescale difference. We found truncated RG (tRG) in ferret cortical development, a progenitor subtype previously described in humans. The combination of in silico and in vivo analyses identified that tRG differentiate into both ependymal and astrogenic cells. Via transcriptomic comparison, we predict that this is also the case in humans. Our findings suggest that tRG plays a role in the formation of adult ventricles, thereby providing the architectural bases for brain expansion.

## eLife assessment

This is an **important** study that improves gene models for the ferret genome and identifies neural progenitors that are comparable to those found in developing human brains. The data are **convincing** and clearly presented. Of particular interest to the field, the work identifies enriched expression of FOXJ1 in late truncated radial glia, strongly indicating that towards the end of neurogenesis, these cells likely give rise to ependymal cells. The work is of interest to anyone studying the development of the nervous system, especially colleagues studying the evolution of development.

## Introduction

A vast diversity of neurons and glia form functional neural circuits during the development of the cerebral cortex in mammals. These cells are progressively generated from multipotent neural stem cells, termed radial glia (RG), following genetic processes common across species, from initially neurons of the deep layers (DL), then neurons of the upper layers (UL), and finally glial cells, astrocytes, or oligodendrocytes (*Figure 1—figure supplement 1A, B*; *Rowitch and Kriegstein, 2010*). These processes

are also spatially organized with RG divisions in the ventricular zone (VZ), IPC-neuron differentiation in the sub-VZ (SVZ), and neuron migration to the cortical plate (CP) in an inside-out manner. In many mammalian phylogenic states, cerebral cortex evolved to gain an additional germinal layer (outer SVZ [OSVZ]; *Smart et al., 2002*), where extensive neurogenesis and gliogenesis of outer RG (oRG) occur (*Hansen et al., 2010*; *Fietz et al., 2010*; *Reillo et al., 2011*; *Gertz and Kriegstein, 2015*), resulting in the amplification of neuronal and glial populations in the cortex (*Figure 1—figure supplement 1*; *Rash et al., 2019*). On the other hand, a new subtype of neural progenitor cell (NPC) in the VZ has recently been reported in humans and rhesus macaques (*deAzevedo et al., 2003*; *Nowakowski et al., 2016*; *Sidman and Rakic, 1973*), lacking the basal attachment and is therefore termed truncated RG (tRG). However, how widely tRG appears in gyrencephalic (or even lissencephalic) mammal development, what mechanisms underlie their formation, and what they produce remain unknown. As distinct NPC subtypes may possess different capacities to generate differentiated progenies (*Huang et al., 2020*; *Rash et al., 2019*), it would be critical to find answers to those questions to understand commonality and diversity in the mammalian brain evolution.

Genetic manipulation of individual cell types in vivo and single-cell transcriptome analysis are two major and successful approaches in revealing the properties of cells, such as their proliferation and differentiation. Thus, single-cell RNA sequencing (scRNA-seq) of the human brain during development has been extensively performed (*Herring et al., 2022*; *Bhaduri et al., 2020*; *Bhaduri et al., 2021*; *Huang et al., 2020*; *Johnson et al., 2015*; *Liu et al., 2017*; *Polioudakis et al., 2019*; *Pollen et al., 2015*; *Zhong et al., 2018*). However, the in vivo behavior in humans and other primates and the underlying mechanisms remain less explored owing to the limited experimental access to the developing primate cortices. Particularly, resources available for the late human embryonic brain are extremely rare due to ethical challenges. Also, studies using brain organoids face issues in recapitulating the specification and maturation of cell types during human brain development (*Bhaduri et al., 2020*). In this context, the ferret (*Mustela putorius furo*) is highlighted as a suitable animal model, which compensates for the difficulties in studying human cortical development. Ferrets are carnivores that develop common gyrencephalic features, such as the OSVZ and a folded brain, and are frequently used in mammalian models of brain development and circuit formation because neurogenesis continues in their early neonatal stages (*Chapman and Stryker, 1992*; *McConnell, 1988*; *Noctor et al., 1997*; *Jackson et al., 1989*). Since this species is experimentally manipulable, recent studies have developed in vivo gene manipulation and editing technology using in utero electroporation (IUE; *Matsui et al., 2013*; *Kawasaki et al., 2012*; *Tsunekawa et al., 2016*; time-lapse imaging; *McConnell, 1995*; *Chenn and McConnell, 1995*). Furthermore, ferrets showed severe microcephalic phenotypes via *ASPM* (*abnormal spindle-like microcephaly-associated* gene) knockout (*Johnson et al., 2018*; *Kou et al., 2015*), which greatly differs from a minor phenotype in mouse *ASPM* knockout mutants (*Capecchi and Pozner, 2015*; *Fujimori et al., 2014*; *Jayaraman et al., 2016*; *Pulvers et al., 2010*). This remarkable finding suggests the presence of mechanisms underlying brain enlargement and circuit complexity shared by gyrencephalic species to some extent. Results from transcriptome profiling of ferret cortical cells have revealed regional differences in germinal layers and cell-type composition (*Johnson et al., 2018*; *de Juan Romero et al., 2015*). However, because of the incomplete genomic information, especially due to the lack of genetic models, the databases with ferret data have been less reliable than those with human or mice data. This has posed a limit for the accurate comparison of single-cell transcriptomes between ferrets and humans. Hence, the temporal pattern of molecular signatures of ferret NPC remains largely unexplored at single-cell resolution. Comparison of progenitor subtypes and sequential events at the single-cell transcriptome level regarding development between ferrets and humans will greatly help to recognize common and species-specific mechanisms underlying the construction of a complex brain.

In this study, we comprehensively analyzed the developmental dynamics of progenitor populations during ferret corticogenesis to clarify shared or species-specific mechanisms of corticogenesis in gyrencephalic mammals and found that ferrets generate tRG at late neurogenic and early gliogenic stages like humans and other primates. Analysis of the pseudo-time trajectory and temporal histochemical pattern suggested that tRG generate ependymal and astrogenic cell fates. Then, we compared temporal series of ferret and human single-cell transcriptomes (*Bhaduri et al., 2021*; *Nowakowski et al., 2017*). Remarkably, we found homologous temporal progenitor trajectories irrespective of the large differential corticogenesis timescale. This study combining rich ferret and human

transcriptome data with in vivo analysis using ferrets emphasizes the value of single-cell ferret transcriptome datasets.

## Results

### Temporal patterns of neurogenesis and gliogenesis in the cerebral cortex of ferrets

First, we confirmed histochemically the spatial and temporal pattern of RG, oRG, and IPC and the appearance of diverse cortical neurons to determine the period in which samples were to be taken for scRNA-seq (*Figure 1—figure supplement 1*). While neurogenesis mostly terminated by P5 in the dorsal cortex (*Figure 1—figure supplement 1A, B*), oligodendrocyte progenitors of dorsal origin (OLIG2⁺) and RG with gliogenic potential (GFAP⁺) became detectable approximately at E40 and increased in number postnatally (*Figure 1—figure supplement 1C, D*). To recapitulate neurogenesis and gliogenesis in the ferret cortex at a high resolution, we decided to determine the temporal trajectories of cell types in the course of ferret brain development (schematically represented in *Figure 1A*) by performing single-cell transcriptomes of neural progenitors, neurons, and glial cells. Based on the above histochemical observations (*Figure 1—figure supplement 1A, B*), we examined single-cell transcriptomes at the six developmental time points: embryonic days E25, E34, and E40 and postnatal days P5 and P10. We carried out scRNA-seq of ferret dorsal cortex at these six developmental time points (*Figure 1*). We prepared two series of cell populations isolated in different ways to enrich the progenitor subtypes (*Figure 1B*): (1) FACS-based sorting of the neural stem cell fraction labeled with an AzamiGreen (AG)-driven *HES5* promoter (*Ohtsuka et al., 2006*) and (2) collecting cells forming the VZ, SVZ, and intermediate zones (IZ) of cerebral cortices after discarding the CP (*Figure 1B*, *Figure 2—figure supplement 1A*).

### Improvement of the gene model for scRNA-seq of ferrets

High-quality information about gene models over the entire genome is prerequisite to obtain a high resolution of single-cell transcriptomes from scRNA-seq. Despite several previous transcriptome studies on the ferret cortex (*Johnson et al., 2018*; *de Juan Romero et al., 2015*), public information on the ferret genome is incomplete; identification of the gene corresponding to a cDNA requires its accurate 3′-untranslated region (3′-UTR) and its correct connection to the coding sequence, because single-cell cDNA library were made by Oligo (dT)-priming method (10x Genomics Chromium, *Figure 1C*). However, information regarding the 3′-UTR of genes had been poor for ferret genomic datasets, which has impeded high-resolution single-cell transcriptome analyses that can offer an accurate comparison with human datasets. Thus, we first improved the annotations of ferret genomic DNA sequences using Chromium droplet sequencing, which tagged all contigs from a long genomic DNA in a droplet, and constructed new gene models based on the improved genomic annotations and newly obtained RNA-seq reads of various tissue types (*Supplementary file 7* and see Materials and methods). Comparing with NCBI references, mapping scRNA-seq reads from E34 to our own references not only improved the mapping rate (number of reads mapped to the transcriptome) from ~30% to ~56%, but also increased the median number of genes detected from ~1000 to ~2000 (*Figure 1D*). We found similar mapping rate and median number of genes detected across all sampling stages (*Supplementary file 1*).

### scRNA-seq revealed subtypes of cortical cells in ferrets

We then combined scRNA-seq information from the two cell populations on all the sampling stages for unbiased clustering. *Figure 2A* shows cell clustering projected in the Uniform Manifold Approximation and Projection (UMAP) space (preprint: *McInnes et al., 2018*; *Stuart et al., 2019*). We characterized 26 transcriptionally distinct clusters from 30,234 ferret cortical cells through corticogenesis and detected up to 2600 median genes per cell (*Figure 1D and E*; *Supplementary file 1*). After combined clustering of the two collectives of cell populations prepared by independent methods, we re-separated cells into each collective, and confirmed a reproducibility of clustering between two different collectives (*Figure 2—figure supplement 1B*).

We also examined whether cells with high mitochondrial contents affect clustering of ferret single cells, because we have decided to include all cells that had less than 30% of mitochondrial genes in

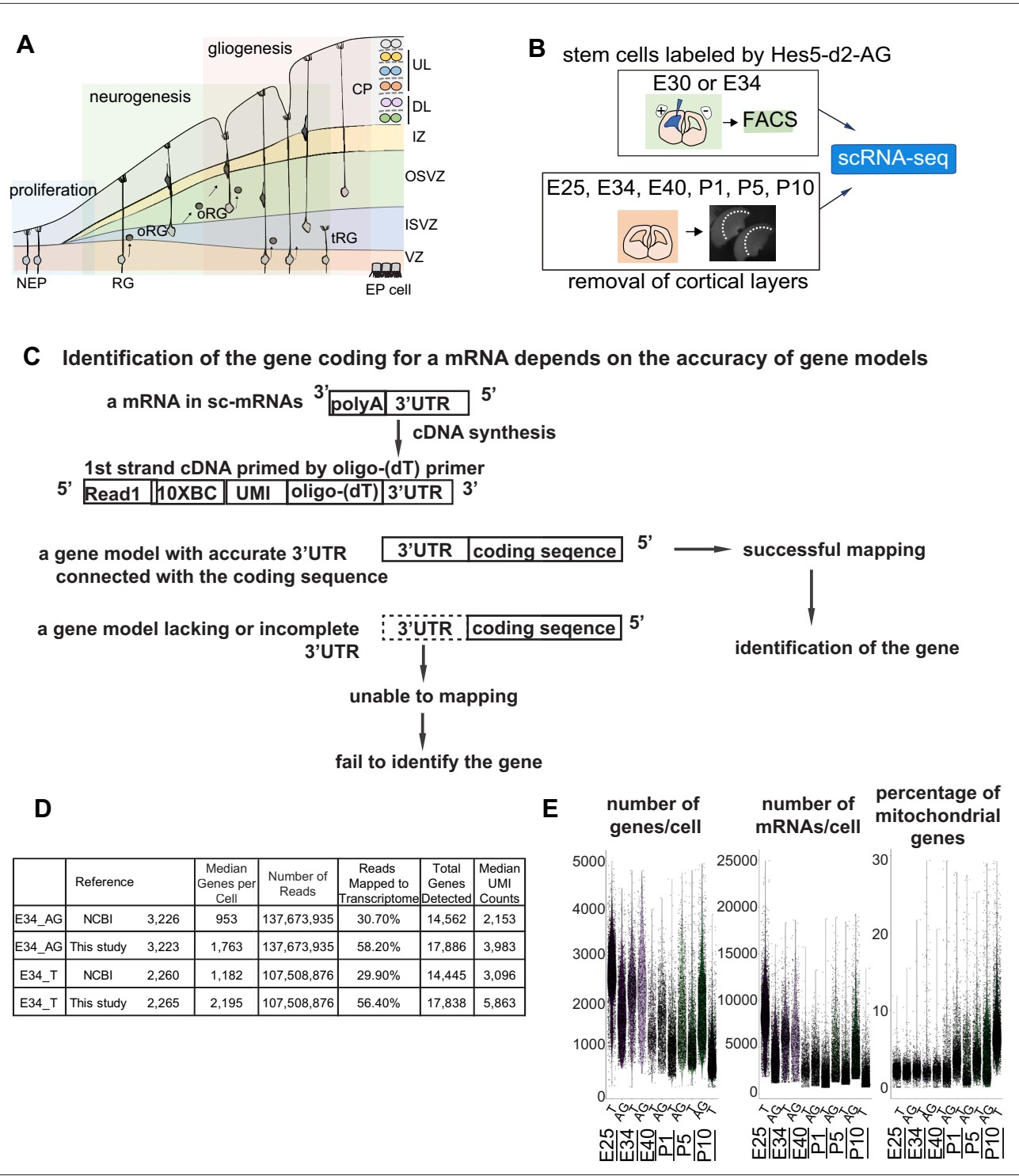

**Figure 1.** Improvement of the gene model for single-cell RNA-sequencing (scRNA-seq) of ferretes. (**A**) Schematic representing cortical development and emergence of diverse neural progenitors in humans and ferrets. Progenitors sequentially generate DL- and subsequent UL-neurons, and finally glial cells, and form ependymal cells. Radial glia (RG) and outer RG (oRG) have been morphologically and positionally identified in both humans and ferrets while truncated RG (tRG) have been only reported in humans. VZ, ventricular zone; ISVZ, inner subventricular zone; OSVZ, outer subventricular zone; IZ, intermediate zone; DL, deep layer; UL, upper layer; CP, cortical plate; NEP cell, neuroepithelial cell; Mn, migrating neuron; and EP cell, ependymal cell. (**B**) Schematic representing the experimental design and time points used to build the transcriptome atlas of developing somatosensory cortex of ferrets. Single cells were isolated using 10x Chromium (see Materials and methods). (**C**) Linkage of the 3'-untranslated region (3'-UTR) sequence and the coding sequence of genes has been improved in this study. This linkage is necessary to assign the gene coding for an mRNA as far as 10x Genomics

*Figure 1 continued on next page*

*Figure 1 continued*

Chromium kit is used for scRNA-seq (see Materials and methods). (**D**) Table comparing quality control metrics of an alignment with either MusPutFur 1.0 (UCSC gene models) or MusPutFur 2.60 (this study) using E34 samples. The total number of genes detected and median genes per cell were higher with MusPutFur 2.60. (**E**) Violin plots showing the number of genes, mRNAs, and the percentage of mitochondrial genes per cell in each sample and time point.

The online version of this article includes the following figure supplement(s) for figure 1:

**Figure supplement 1.** Temporal pattern of neurogenesis and gliogenesis in the cerebral cortex of ferrets.

our analysis based on the percentage of reads mapped on the mitochondrial genome; we found that the majority of cells in each cell type had a value less than 5% while some cells contained them in the range between 0% and 10%, up to a maximum of 28% (*Figure 1E*). We confirmed that single cells after filtering cells with the threshold of 10% mitochondrial content (28,686 cells in our dataset) and those with the threshold of 30% (*Figure 2A*) provided similar clustering patterns with each other, both of which 26 clusters (*Figure 2—figure supplement 1C*) indicating that the current selection of cells with the mitochondrial contents is appropriate.

Cell clusters were annotated according to their specific gene expression patterns (*Figure 2B and C*, *Figure 2—figure supplement 2A*, *Supplementary file 1*) and assigned into 10 cell types: RG (early, mid, and late), IPC, OPC, ependymal cells, excitatory cortical neurons (DL and UL), inhibitory neurons (ITN), microglia, endothelial cells, mural cells, and unknown cells (*Figure 2A*). RG, IPC, and neuronal clusters were aligned according to the neuronal differentiation process in the UMAP plot (*Figure 2A*). RG cells were classified into three clusters (early, mid, and late) according to their collection stages and also the expression of temporally altered RG markers reported previously (hereafter named temporal markers) (*Okamoto et al., 2016*; *Telley et al., 2019*). The 'early RG' clusters comprised E25 cells while the 'mid RG' clusters, mostly E34 cells (*Figure 2—figure supplement 2B*). Early and late RG, and IPC were subdivided into three subclusters that expressed different cell cycle markers (*Figure 2—figure supplement 2C*).

Remarkably, we identified a small cluster (409 cells) of *PAX6*-expressing RG subtype, 69% of which expressed *CRYAB* (*Figure 2D*). *CRYAB*, encoding a molecular chaperone (*Yamamoto et al., 2014*), is a unique marker for human tRG (*Nowakowski et al., 2016*) therefore, we designated them as tRG-like cells in ferrets. In contrast, oRG cells failed to be distinguished from ventricular RG cells (vRG) in ferrets by unbiased clustering alone (see below). HOPX, a typical marker for oRG in human tissues (*Figure 2E*, *Figure 2—figure supplement 2D*; *Pollen et al., 2015*), has been indeed detected in both oRG and vRG at late stages in vivo in ferrets (*Figure 2—figure supplement 2D*; *Johnson et al., 2018*; *Kawaue et al., 2019*). tRG emerge around birth during the development of somatosensory cortex in ferrets.

The presence of a tRG-like cluster in ferret transcriptomes led us to investigate the properties of the cells in this cluster. To examine their cell morphology, we sparsely labeled NPC by electroporating P0 embryos with an expression vector for enhanced green fluorescent protein (EGFP) and found a cell population showing major characteristics of human tRG cells (*Nowakowski et al., 2016*): expression of CRYAB and PAX6 with an apical endfoot and truncated basal fiber in the VZ during late neurogenesis (*Figure 3A and B*, *Figure 3—videos 1 and 2*). CRYAB expression emerged shortly prior to birth (day 41), gradually increasing in cell number, and became mostly restricted to tRG-shaped cells in the VZ and SVZ (*Figure 3C and D*, *Figure 3—figure supplement 1A*). These CRYAB⁺ cells were mostly post-mitotic (KI67-) and neither IPC (TBR2+) nor OPC (OLIG2+) at P10, while a small fraction of those cells expressed these markers at P5 (*Figure 3E–G*). These histochemical observations were consistent with our single-cell transcriptome data (*Figure 3—figure supplement 1A–F*). Altogether, we concluded that tRG-like cells in ferrets are equivalent to human tRG cells (as also confirmed by transcriptomic comparison between humans and ferrets, shown later). A difference in the late neurogenic cortex between humans and ferrets is that the VZ and OSVZ are connected by conventional RG cells in ferrets (*Figure 3A*, *Figure 3—figure supplement 1G*), but separated in humans due to the disappearance of conventional RG cells (*Nowakowski et al., 2016*).

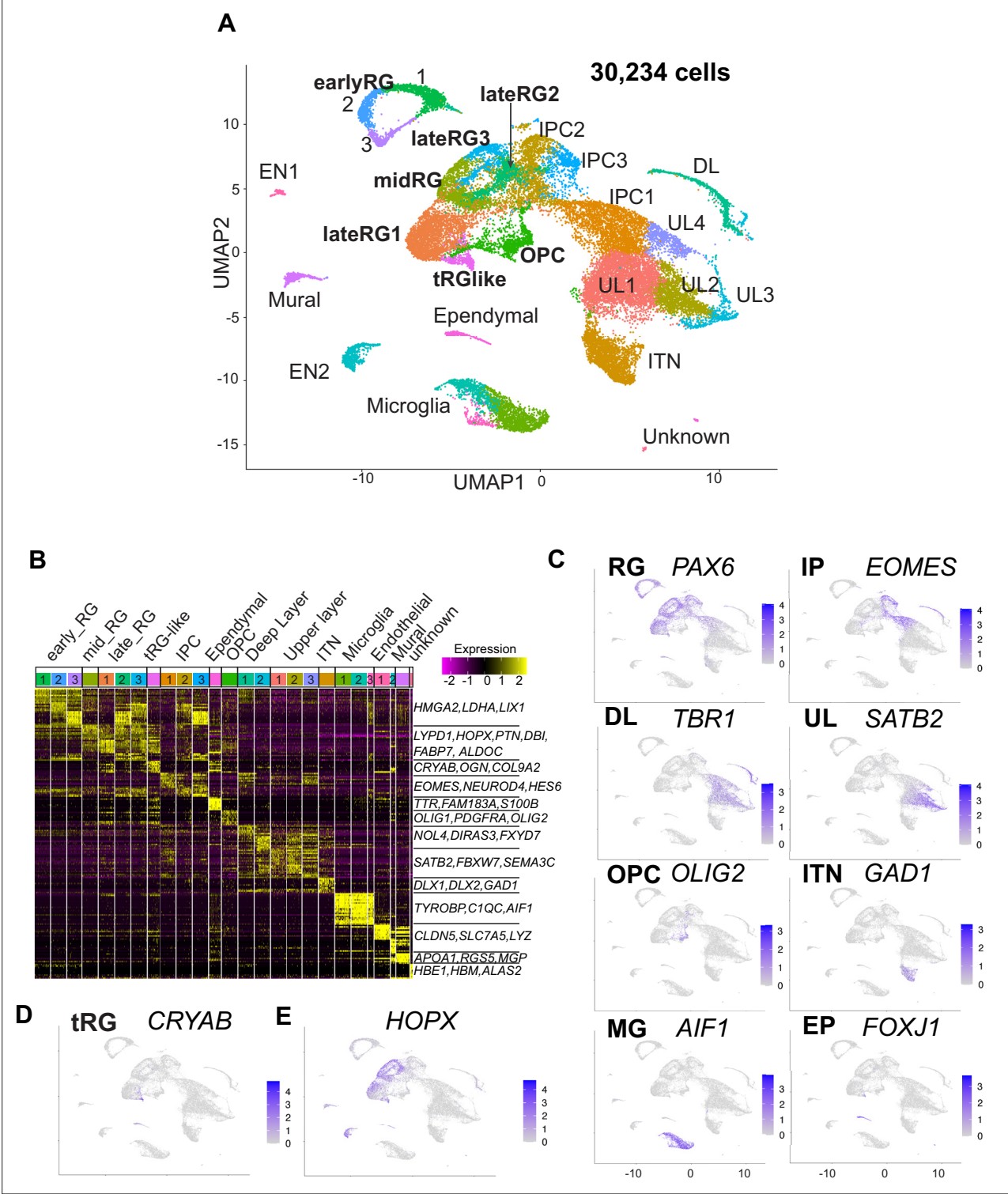

**Figure 2.** Single-cell RNA-sequencing reveals ferret transcriptome signatures and the cell types. (**A**) Uniform Manifold Approximation and Projection (UMAP) visualization showing cells colored by Seurat clusters and annotated by cell types (as shown in **C**, **D**). (**B**) Heatmap showing expression profiles of cluster marker genes based on log fold change values. Cells were grouped by Seurat clustering (transverse). Cell types were assigned according to the expression of marker and differentially expressed genes in each cluster. Early radial glia (RG) was defined by the expression of *HMGA2, LDHA*, and *LIX1*; and late RG, by *PTN, ALDOC*, and *FABP7*. OPC, oligodendrocyte precursor cell; ITN, interneuron; and EN, endothelial cells. Other cell type abbreviations are shown in **A** and the main text. Color bar matches the Seurat clusters in (**A**). The 10 most enriched representative genes in each cluster are shown, with typical marker genes noted in (**C–E**). (**C**) Normalized expression levels of representative marker genes of different cell types projected

*Figure 2 continued on next page*

*Figure 2 continued*

onto the UMAP plot in (**A**). MG, microglia. (**D**) Expression pattern of *CRYAB*, a marker for truncated RG (tRG), only described in humans and other primates. (**E**) The expression pattern of *HOPX*, a marker for outer RG (oRG) in humans and other primates in the UMAP plot.

The online version of this article includes the following figure supplement(s) for figure 2:

**Figure supplement 1.** Quality and reproducibility of ferret single-cell transcriptome dataset.

**Figure supplement 2.** Cell types in developing ferret cerebral cortex.

## Temporal fates of RG cells are predicted by pseudo-time trajectory analysis

To understand the relationship between various cortical progenitors in the developing ferret brain, particularly in the origin and fate of ferret tRG, pseudo-time trajectory analysis was performed. Single cells that had been subjected to single-cell transcriptome analyses (from E25 to P10) were unbiasedly ordered along a trajectory based on their transcriptome profiles (*Figure 4—figure supplement 1A, B*). To simplify our analysis, we first excluded interneurons, microglia, endothelial cells, and excitatory neuronal clusters. Subsequently, 6000 single cells were randomly selected from the remaining cell population for further analysis. The pseudo-time analysis predicted a reasonable trajectory consisting of three branching points that generated seven states (*Figure 4A*, *Figure 4—figure supplement 1B*). We assigned major cell types for each state based on the cell clusters defined by UMAP analysis (*Figure 3A*, *Supplementary file 2*). The trajectory successfully predicted cortical development, along which *HES1*+ stem cells shift their features from the earlier to the later stages, generating branches of differentiated cells (*Figure 4B and C*, *Figure 4—figure supplement 1D*). Major *HES1*+ stem cell states (states 1, 3, 5, and 7) contained cells at different stages; thus, termed NPC1, -2, -3 and astrogenic, respectively (*Figure 4—figure supplement 1C*). NPC1 (mostly from E25) bifurcated into the neuronally differentiated lineage and NPC2 state (*Figure 4—figure supplement 1B*), which produced the OPC lineage and NPC3 state at the second branching point. When neurogenesis gradually declined after birth, the NPC3 state, mainly consisting of late RG cells and tRG (*Figure 4—figure supplement 1C*), adopted two states, state 6 with an presumptive ependymal fate (*FOXJ1*+, *SPARCL1*+) and state 7 with presumptive astrogenic fate (*FOXJ1*-, *SPARCL1*+), as judged by the combination of late fate markers (*Liu et al., 2022*; *Saadoun and Papadopoulos, 2010*; *Yu et al., 2008*, *Figure 4D*, *Figure 2—figure supplement 2A*).

Remarkably, tRG cell population was distributed into three branches along the pseudo-time trajectory (*Figure 4E and F*): NPC3 (mainly during E40–P1), astrogenic (P1–P10), and ependymal (P5–P10) branches, suggesting that tRG prenatally arise as precursors of ependyma and astroglia in the ferret cortex (see below). Different territories were assigned into these three states in the UMAP even when we examined the entire tRG population (409 cells) before making a random selection of 6000 cells for pseudo-time trajectory analysis, supporting the reliability of this presumptive categorization of tRG (*Figure 4E*).

## tRG cells are likely to possess ependymal and gliogenic potential during cortical development in ferrets

To test the hypothesis that tRG generate both ependymal cells and astroglia populations, we first examined the expression of an ependymal marker, FOXJ1, a master regulator of ciliogenesis, according to CRYAB expression in the VZ. We observed that CRYAB+ tRG cells gradually co-expressed FOXJ1, reaching up to 90% co-expression by P10 (*Figure 5A and B*). Similarly, our transcriptome data showed that the fraction of *CRYAB-FOXJ1* double-positive cells increased in the tRG cluster from P1 to P10, whereas other RG clusters maintained low *FOXJ1* expression (*Figure 5C and D*). From P5 onward, differentiating ependymal cells (*CRYAB-FOXJ1* double-positive) accumulated within the VZ (*Figure 5A*). These cells often align their nuclei in parallel to the ventricular surface, near which nuclear-lined aggregates are observed more frequently (between two arrowheads in P5 and P10 in *Figure 5A*), and their cell body finally settled on the apical surface with short basal fibers by P14 (*Figure 5E*). These data suggest that a majority of tRG cells progressively upregulates *FOXJ1* expression to adopt an ependymal cell fate during postnatal development. Concomitantly, adenine cyclase III expression in both primary and multiciliated cells indicated that ciliogenesis progressed postnatally, forming multi-ciliated ependymal cells on the ventricular surface of the ferret cortex by

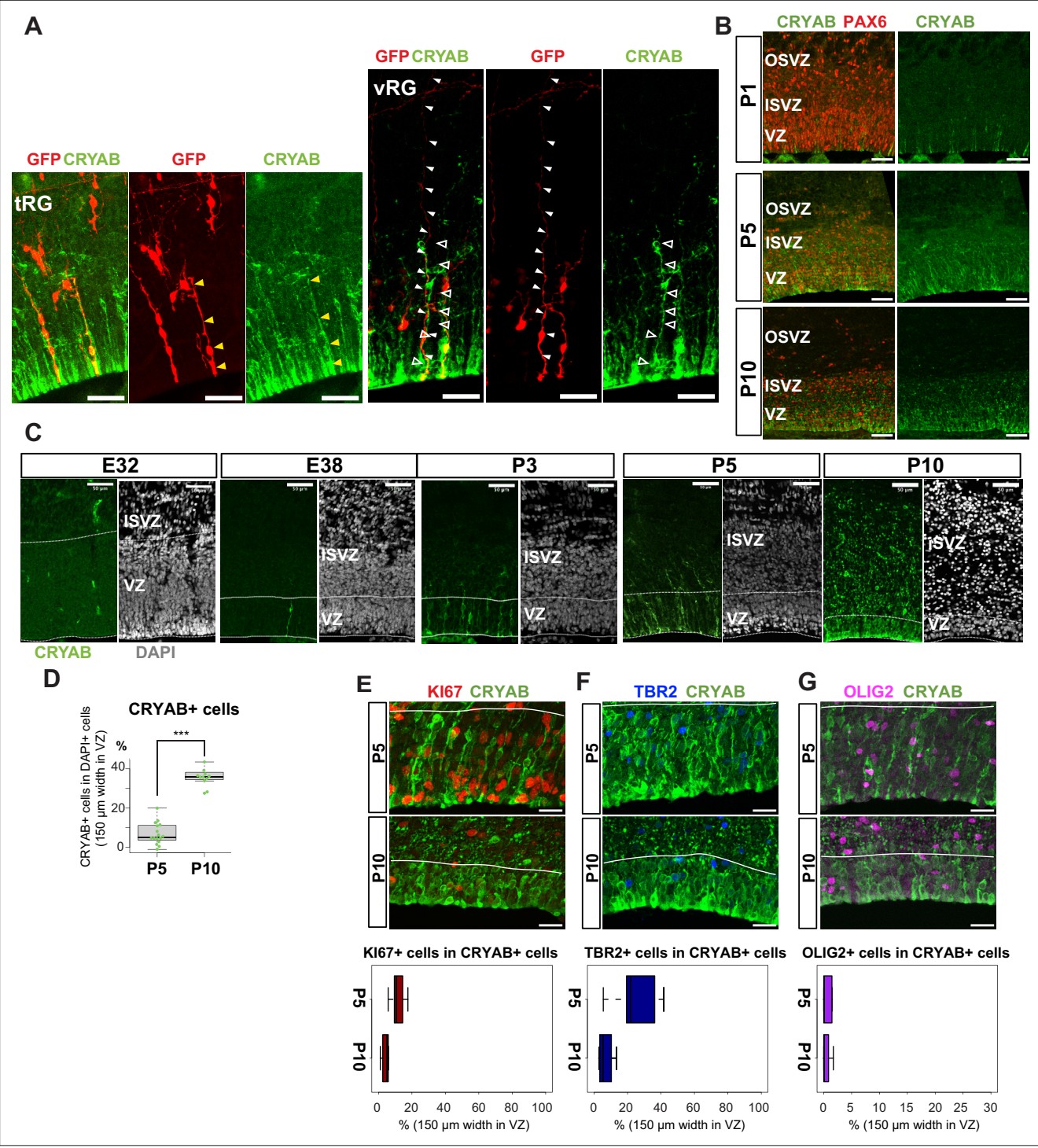

**Figure 3.** Truncated truncated radial glia (tRG) emerges during postnatal cortical development in ferrets. (**A**) Representative images showing the cellular features of tRG and aRG in ventricular zone (VZ) at P0 and P5, respectively, stained for GFP (red) and CRYAB (green). RG cells were sparsely labeled with a GFP-expressing plasmid at E30 via in utero electroporation (IUE) for P0 samples, and at P3 for P5 samples. MAX projection was performed on a 30 μm vibratome section with 5 μm or 2.5 μm interval for each z-image at P0 or P5, respectively. Scale bar, 50 μm. (**B**) Expression patterns of CRYAB (green) and PAX6 (red) in ferret germinal zones during postnatal development (P1, P5, and P10) (cryosection thickness = 12 μm). (**C**) Developmental profile of CRYAB expression in ferret cortices. Immunostaining for CRYAB (green) and 4′,6-diamidino-2-phenylindole (DAPI) (gray) on cryosections at E32, E38, P3, P5, and P10 (cryosection thickness = 12 μm). (**D**) Quantification of CRYAB$^+$ cells among all nuclei (DAPI) in VZ strips through immunostaining, indicating a great increase in CRYAB-expressing cells (strip width = 150 μm; n=3 for P5; n=2 for P10; Wilcoxon rank-sum test, p-value=6.574e-08). (**E–G**) Expression of other markers in CRYAB$^+$ cells. Representative images taken with a 100× objective are shown with MAX projection. The border of VZ is shown with a white line.

*Figure 3 continued on next page*

*Figure 3 continued*

Below each double staining image, quantification of KI67-, TBR2-, or OLIG2-expressing cells among the CRYAB⁺ cell population in the VZ is shown (n=2 for P5; n=2 for P10 for each staining except n=3 for TBR2 staining). Box and whisker plots indicate ranges (lines) and upper and lower quartiles (box) with the median. Scale bars, 20 µm.

The online version of this article includes the following video and figure supplement(s) for figure 3:

**Figure supplement 1.** Development of ferret truncated radial glia (tRG) during postnatal cortical development.

**Figure 3—video 1.** Rotated 3D-reconstructed image of the transparent slice used for taking the images in *Figure 3A*.

https://elifesciences.org/articles/91406/figures#fig3video1

**Figure 3—video 2.** Rotated 3D-reconstructed image of the slice, from which the image of *Figure 3—figure supplement 1G* was taken.

https://elifesciences.org/articles/91406/figures#fig3video2

P35 (*Figure 5—figure supplement 1A*). Altogether, it is most likely that FOXJ1⁺tRG are fated to be ependymal cells that constitute the ventricular surface. To prove this, it is necessary to chase tRG by knocking in a fluorescent marker gene via IUE, or to follow the change in cell shape from tRG to a characteristic ependymal morphology in slice cultures by labeling with a fluorescent gene via IUE. However, the latter would not be realistic, because the process of the transformation from tRG to ependymal cells is slow to take a long time in an order of 10 days. It is an important future issue to develop the strategy to genetically follow the fate of tRG.

Moreover, a fraction of tRG belongs to the state 7 (astrogenic state) in the pseudo-time trajectory (*Figure 4E and F*). We examined whether these tRG cells differentiate into astrogenic cells. We chose *FOXJ1* as an ependymal marker and *SPARCL1* as an early astroglial marker based on a recent study in mice (*Liu et al., 2022*; *Figure 5—figure supplement 1B*). While *SPARCL1* is expressed in all states generated by NPC3, *SPARCL1* expression is highest in astrogenic tRG among the three tRG subgroups, whereas *FOXJ1* expression is nearly exclusive to ependymal tRG (*Figure 5F*). Furthermore, cells expressing high levels of *SPARCL1* do not express *FOXJ1* (*Figure 5—figure supplement 1C*, *Supplementary file 3*). These features of the astrogenic tRG are essentially the same as those of the astrogenic state (state 7) of the pseudo-time trajectory (*Figure 4D and E*, *Figure 5G*). Consistently, whereas FOXJ1⁺ cells were located close to the apical surface, those expressing *SPARCL1* were observed relatively far from the apical surface in the VZ (*Figure 5H, I*, *Figure 5—figure supplement 1D*, *Supplementary file 4*). Taken together, our analyses strongly suggest that tRG cells adopt both the ependymal and astroglial fates.

## Ferrets and humans show a homologous developmental transcriptomic profile of progenitor subtypes

Both ferrets and humans are positioned in distinct phylogenetic branches; both represent complex brain features of gyrencephalic mammals. Therefore, we compared our temporal series of ferret single-cell transcriptomes with a previously published human dataset (*Nowakowski et al., 2017*) to examine which processes are species-specific or common for the two species. We merged the two datasets by pairing mutual nearest neighbor cells (MNN) following canonical correlation analysis (CCA) across species (*Stuart et al., 2019*) and found that many cell types, including RG, IPC, OPC, and neurons, were clustered together (*Figure 6A*, *Figure 6—figure supplement 1A*). Temporal patterns and variety of neural progenitors during cortical developments were similar to each other between humans and ferrets at the single-cell transcriptome level (*Figure 6—figure supplement 1B*). Nonetheless, the timescales of cortical development significantly differed: on the UMAP plot, ferret E25 cells closely distributed with human GW8 cells; ferret E34, with human GW11–14; ferret E40–P1, with human GW15–16; and ferret P5–P10, with human GW17–22 (*Figure 6B*).

Next, we compared the similarities among individual RG subtypes across species. To quantify the features of an RG subtype, we introduced the parameter 'genescore' (*Bhaduri et al., 2020*), a parameter that reflects the enrichment and specificity of a marker gene for a given cluster; the genescore for a particular marker gene is defined by multiplying the average enrichment of expression quantity in cells of the cluster (fold change) and the ratio of the number of marker-expressing cells in the cluster to the cell number in all other clusters (see Materials and methods). Comparison of genescores of two arbitrary RG subtypes between two species revealed a high correlation in the early and late RG clusters (vRG and oRG for the human dataset) across species (*Figure 6C and D*, *Supplementary file 5*).

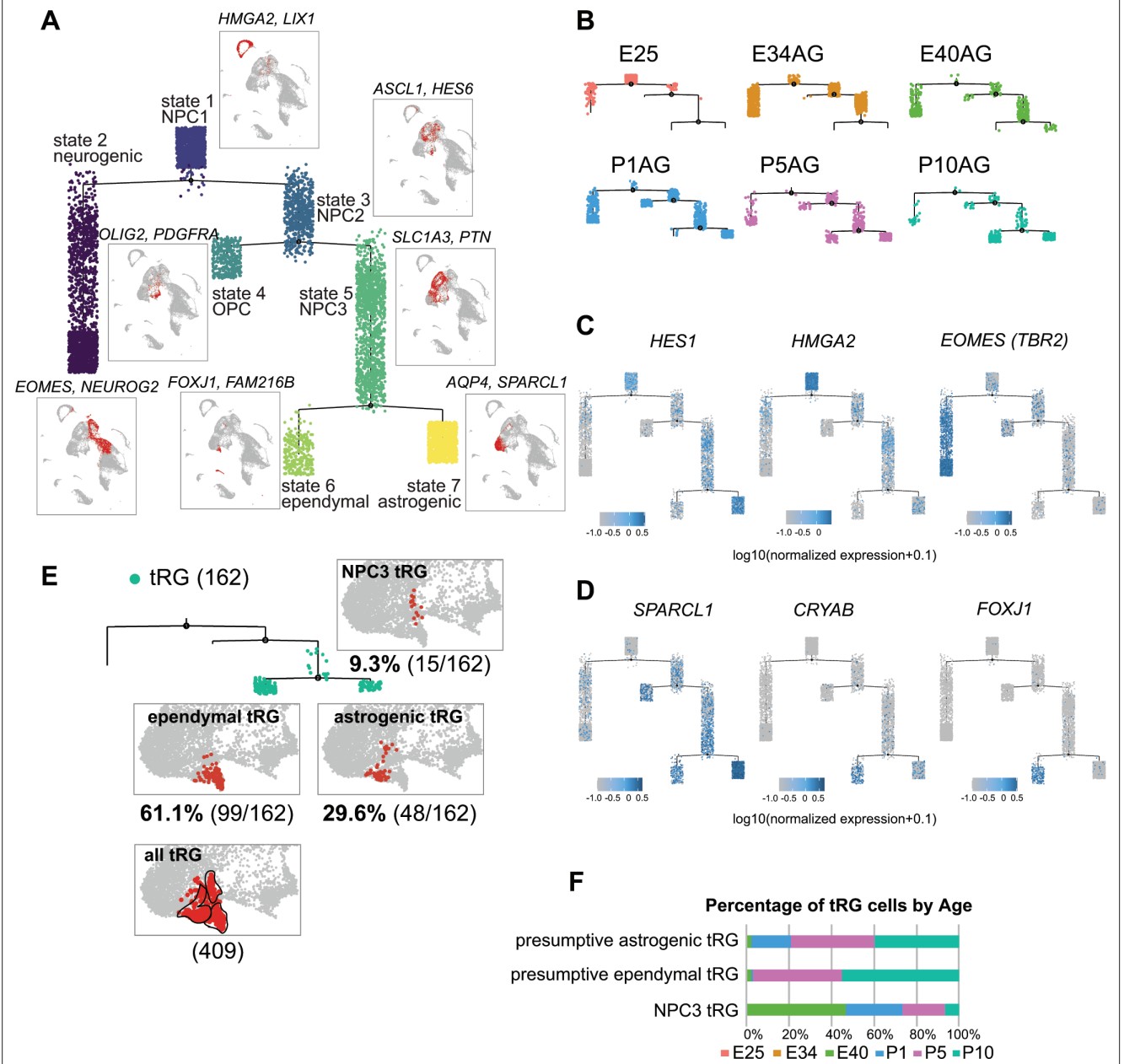

**Figure 4.** Temporal fates of radial glia (RG) cells predicted by pseudo-time trajectory analysis. (**A**) Pseudo-time trajectory tree of progenitor cells in the ferret developing cortex (assembled with Monocle v2). Cellular distribution at each state is the same as in the Uniform Manifold Approximation and Projection (UMAP) plot shown in *Figure 1C*. Cell types representing each state and their gene markers are shown next to each state (see below). (**B**) Trajectory trees split by collection stages (AzamiGreen [AG] samples are shown). (**C**) Distribution of cells expressing marker genes for stem cell states (*HES1* and *HMGA2*) and the neurogenic state (*EOMES*) along trajectories. Color densities indicate the log-normalized unique molecular identifier count for each gene. (**D**) Normalized expression of three genes: *SPARCL1*, *CRYAB*, and *FOXJ1* in the state 6 and state 7 cells in the pseudo-time trajectory tree (**A**). The combination of expression levels of the four marker genes discriminates state 6 (ependymal) and state 7 (astrogenic). See the text for details. (**E**) Distribution of truncated RG (tRG) cells along the tree and tRG-focused UMAP visualization. The tRG cells (n=162) were found on the three states, NPC3 (9.3%; 15 cells), ependymal tRG (61.1%; 99 cells), and astrogenic tRG (29.6%; 48 cells). (**F**) Composition of the three types of tRG in (**D**) by the collection stage (AG and T samples are combined). No tRG cells were collected from E25 and E34.

The online version of this article includes the following figure supplement(s) for figure 4:

**Figure supplement 1.** Temporal fates of radial glia (RG) cells identified by pseudo-time trajectory analysis.

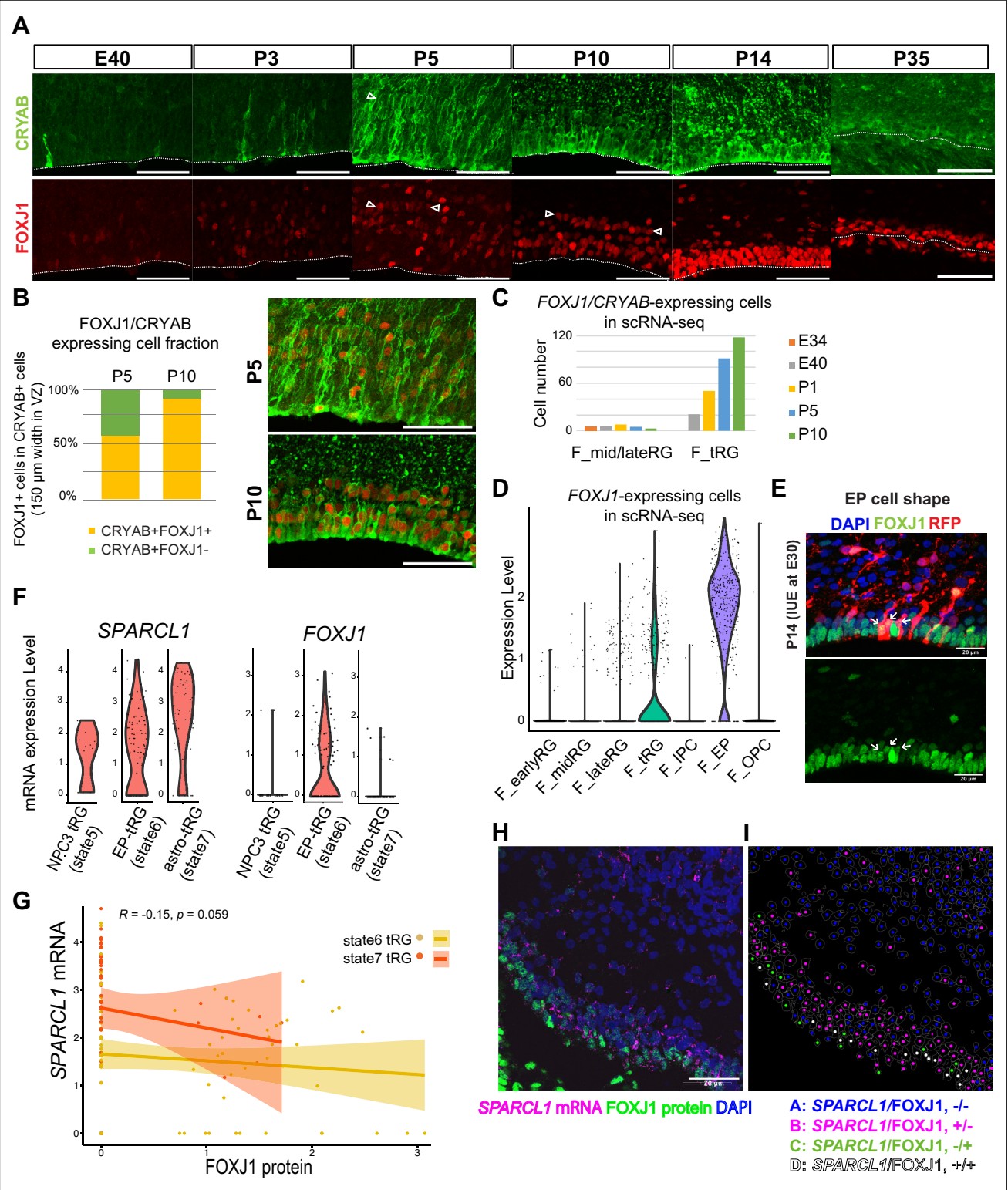

**Figure 5.** Truncated radial glia (tRG) adopt both ependymal cells and astrogenic fates in ferrets. (**A**) Immunostaining of 12 µm cortical cryosections with CRYAB (green) and FOXJ1 (red), focusing on the ventricular zone (VZ) at E40, P3, P5, P10, P14, and P35. Scale bar, 50 µm. CRYAB and FOXJ1 double-positive nuclear rows are shown with arrowheads at P5 and P10 when this nuclear alignment is visible. (**B**) Percentage of FOXJ1 expression in CRYAB-expressing cells (n=3 for P5; n=2 for P10; Wilcoxon rank-sum test p-value=1.749e-05). Images with merged channels in A are shown with the same color codes, antibodies, and scale bars as A. (**C**) Number of cells expressing both *CRYAB* and *FOXJ1* in the mid and late RG, or tRG clusters shown in *Figure 2A* (with unique molecular identifier counts higher than 0.25). Colored bars indicate the stages of sample collection. *CRYAB*- and *FOXJ1*-

*Figure 5 continued on next page*

*Figure 5 continued*

expressing cells increased over time only in the tRG cluster. (**D**) Normalized expression levels of *FOXJ1* in each cell in the indicated ferret clusters in *Figure 2A*. (**E**) Cortical origin and shape of FOXJ1-expressing ependymal cells indicated with white arrows. Staining for FOXJ1 at P14 after labeling cortical progenitors with an mCherry-expressing vector via in utero electroporation (IUE) at E30. The maximum projection images with 1 μm z-step size are shown. Cryosection thickness = 12 μm; scale bar = 20 μm. (**F**) Normalized expression of *SPARCL1* and *FOXJ1* in each cell in ferret tRG clusters separated by the pseudo-time trajectory analysis (see *Supplementary file 3*). (**G**) Relationship between *SPARCL1* and *FOXJ1* transcripts in individual cells within each tRG cluster. Within the 15 cells that were classified in state 5 tRG, only one expressed *FOXJ1* mRNA. Pearson relationship analysis was performed using cells from states 6 and 7 (R=0.19, p-value = 0.018). (**H**) Immunostaining of a P10 cortical tissue for FOXJ1 protein and S*PARCL1* mRNA. (**I**) Cells within the VZ in (**H**) were clustered into four classes by FOXJ1 protein± and S*PARCL1* mRNA±. Cluster A: S*PARCL1*/FOXJ1-/-, B: S*PARCL1*/ FOXJ1+/-, C: S*PARCL1*/FOXJ1-/+, D: S*PARCL1* /FOXJ1+/+ (*Supplementary file 4*). See Materials and methods for clustering. Scale bar = 20 μm.

The online version of this article includes the following figure supplement(s) for figure 5:

**Figure supplement 1.** Ferret truncated radial glia (tRG) adopt both ependymal and astrogenic fates in ferrets.

tRG cells also showed a remarkable similarity between the two species (*Figure 6C and D*), as represented by a high level of expression for the combination of *CRYAB*, *EGR1*, and *CYR61* (*Figure 6E*). To better examine this similarity, we defined the 'cluster score' for individual cells as a linear combination of the expression level of marker genes in a cell, which is weighted by its gene-score in the human cluster of interest. Calculation of the cluster score for ferret tRG using gene-scores of the human tRG cluster resulted in a higher score than any other cluster in both datasets (*Figure 6F*). These results confirm, via transcriptomics, that the ferret tRG cluster is very close to the human one.

## Transcriptional analysis of human tRG subtypes by integration with ferret tRG subtypes

Next, we assessed whether human tRG cells also possessed ependymal and gliogenic potential, as suggested in ferret cells. We chose a recently published human dataset (*Bhaduri et al., 2021*) for comparison, because this study containing GW25 dataset which included more tRG cells than previous studies did not contain GW25 data. Furthermore, we used only data at GW25. After excluding neurons and other cell types from the analysis, we identified respective human clusters for tRG and oRG cells based on their marker gene expression (*Figure 7A*, *Supplementary file 6*). Then, we merged this human dataset and ferret NPCs (*Figure 7B and C*) by the MNN after CCA as described before when we merged the entire series of progenitor samples of ferrets and humans (*Figure 6A*). This procedure revealed that human tRG and oRG share similar transcriptomes with ferret tRG and late RG that must include oRG, respectively (*Figure 7D–H* and *Figure 7—figure supplement 1A, B*). It is worth noting that early RG in ferrets showed the highest similarity with 'OLIG1' in human data, and no cell type in human data corresponded to ferret 'midRG', likely because only GW25 cells were used for comparison (*Figure 7—figure supplement 1A, B*).

The integration of ferret and human datasets showed that tRG from both species were mainly distributed into three different clusters in the UMAP space: 7, 21, and 28 (*Figure 7F*, *Figure 7—figure supplement 1C*). Cluster 7 cells highly expressed late onset RG genes (such as *APOE*, *FABP7*, *NOTCH2*, and *DBI*; *Figure 7—figure supplement 1D*), suggesting that tRG cells in cluster 7 were presumably at a late RG-like state, being not yet committed to neither astrogenic nor ependymal fates (mostly in the NPC 3 state in the ferret pseudo-time trajectory; *Figure 4A and E*). In contrast, cells in clusters 21 and 28 highly expressed marker genes for the astrocyte (*GFAP* and *AQP4*), and the ependymal (*FOXJ1*) clusters, respectively (*Figure 7—figure supplement 1D*, *Supplementary file 6*). Then, we confirmed that the three states of ferret tRG cells, committed to the astroglia, ependymal fates, and uncommitted states, were largely assigned to clusters 21, 28, and 7, respectively (*Figure 7G*), guaranteeing the reliability of this method. Thus, our results raise the possibility that tRG cells in humans possess the potential to generate astrocytes and ependymal cells as suggested for ferret tRG. This analysis also reveals that the proportion of the three tRG subtypes was very different between human and ferret datasets (*Figure 7F*); the two presumptively committed tRG are very minor populations in humans, possibly due to the differences in developmental stages where tRG of these subtypes are enriched (see Discussion).

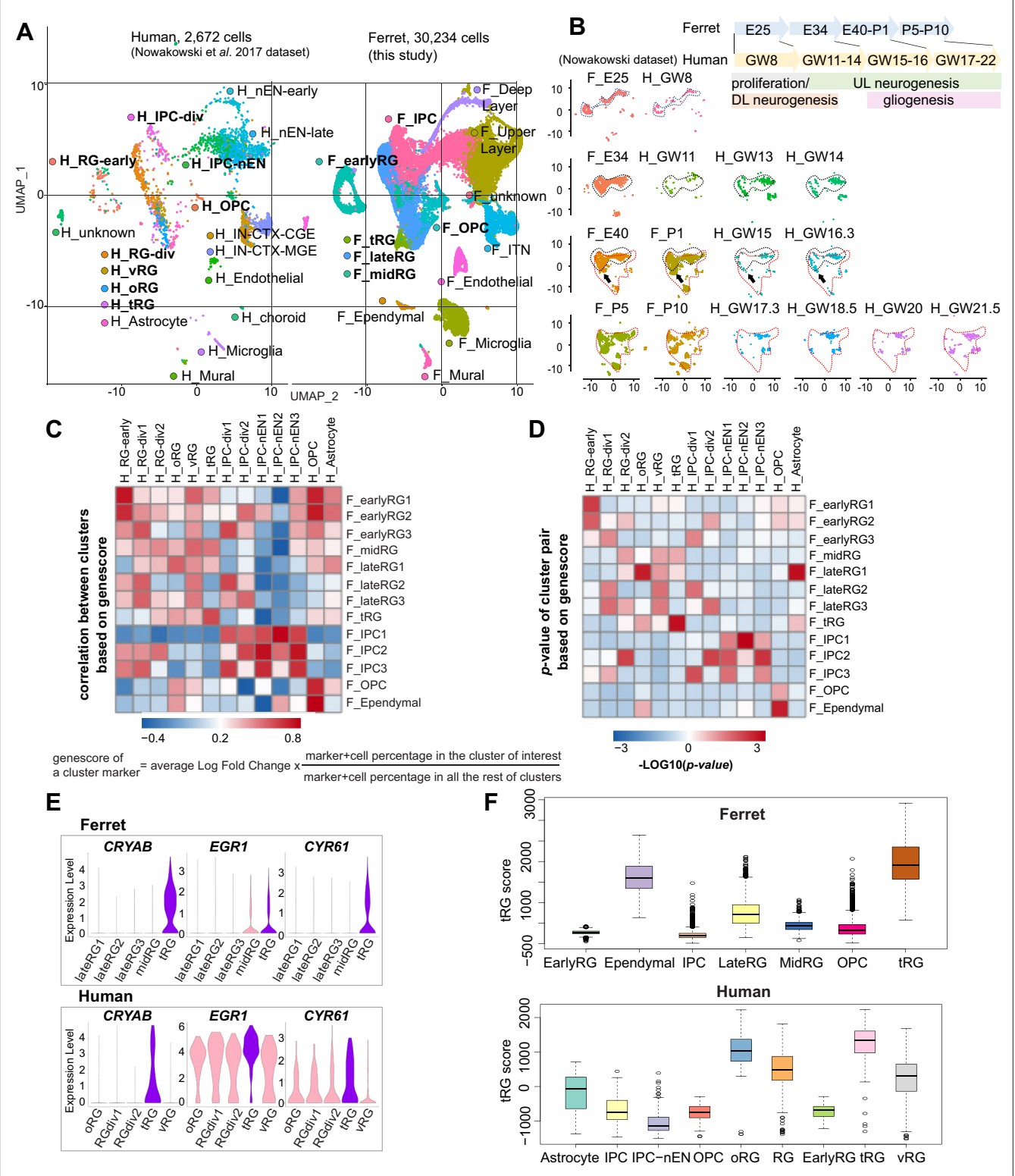

**Figure 6.** Comparison of molecular identity of radial glia (RG) subtypes between ferret and human. (**A**) Uniform Manifold Approximation and Projection (UMAP) visualization of integrated human (n=2672; left) and ferret (n=30,234; right) single-cell datasets colored according to the different clusters. The names of clusters from human and ferret cells begin with 'H' and 'F', respectively. (**B**) Homologous temporal pattern of transcriptomic characters of progenitors and their progenies between ferrets and humans. Corresponding neurogenic stages between ferrets and humans can be assigned by cell distribution at homologous positions in UMAP plots. Gliogenic RG cells (a subtype of the 'late_RG' group and OPC in **Figure 2A**)

*Figure 6 continued on next page*

*Figure 6 continued*

were first distinguished transcriptionally at ferret E40 and at human GW14, as pointed out by arrows. (**C, D**) Correlation coefficient (**C**) and significance (**D**) between indicated clusters of ferrets and humans, calculated by marker gene scores. See the text and Materials and methods for details. (**E**) Normalized expression levels of the indicated genes in humans and ferrets from each progenitor cluster. See the text and Materials and methods for details. (**F**) Truncated RG (tRG) scores of the indicated clusters are presented as box and whisker plots for humans and ferrets. As for the definition of cluster scores, see the text and Material and methods. These represent the range without outliers (lines) and upper and lower quartiles with the median (box). Outliers are represented as points outside the box. Outliers are 1.5-fold larger or smaller than interquartile range from the third or first quartile, respectively.

The online version of this article includes the following figure supplement(s) for figure 6:

**Figure supplement 1.** Comparison of molecular identity of radial glia (RG) subtypes between ferret and human.

## Human cortical organoids lack tRG population

Human cortical organoids are a powerful model for understanding cortical development (*Eiraku et al., 2008*). It is, therefore, important to know how developing ferret cortices resemble human cortical organoids especially regarding to features shared by human tissues and ferrets. We then decided to analyze two different datasets of cortical organoids to see whether tRG is present in those organoids (*Figure 7—figure supplement 2A–C* for *Bhaduri et al., 2020*; *Figure 7—figure supplement 2D* for *Herring et al., 2022*), by focusing on CRYAB-expressing cells. We assigned cells according to the original annotations, which included the information of organoid lines, age, and state (*Figure 7—figure supplement 2A*). Our clustering resulted in 32 clusters where CRYAB-expressing cells were detected in two out of four cell lines (*Figure 7—figure supplement 2B*). We noted that a tRG cluster was absent in the original annotation. To further confirm the absence of a tRG cluster in organoids, we integrated organoid dataset with the dataset derived from human primary tissues (*Bhaduri et al., 2020*). After the integration, we failed to find organoid-derived CRYAB-expressing cells that overlapped with tRG cells from the human primary tissues (*Bhaduri et al., 2020*; *Figure 7—figure supplement 2C*), nor CRYAB-expressing cell itself in the other dataset (*Herring et al., 2022*; *Figure 7—figure supplement 2D*). Our analyses thus indicate that tRG-like populations seem to be lacking in organoid datasets that are currently available (*Bhaduri et al., 2020*; *Herring et al., 2022*).

## Prediction of ferret oRG-like cells via the identification of cells homologous to human oRG cells

As mentioned before, oRG could not be identified as a separate cluster in the ferret dataset (*Figure 2*). We attempted to assign oRG-like cells in the ferret dataset using human oRG cells as anchors via the integration of two datasets by the MNN after CCA (*Figure 7B*). We could assign oRG-like cells, which were located near the human oRG cluster (*Figure 7H*). To assess the degree of similarity to human oRG, we calculated the oRG score for each ferret oRG-like cell as we did for tRG (*Figure 6F*). The assigned oRG-like cells had significantly higher oRG cluster scores than all other NPCs (*Figure 7I*). Furthermore, we searched genes that were highly expressed in oRG-like cells by comparing oRG-like cells and other NPCs in the ferret dataset (*Supplementary file 6*). Consistent with the human dataset (*Bhaduri et al., 2021*), the expression of *HOPX, CLU,* and *CRYM* was higher in oRG-like cells than in other ferret NPCs (*Figure 7J*, *Figure 7—figure supplement 1E*). However, human oRG markers such as *HOPX* and *CLU* were also expressed by vRG and tRG in ferrets (*Figure 2—figure supplement 2D*, *Figure 7—figure supplement 1F*). Therefore, oRG at the transcriptome level can only be distinguished by the similarity with the whole transcriptome or a combination of several markers, but not by a few marker genes, whereas oRG can be unambiguously identified by its location in the tissue (*Figure 1—figure supplement 1*).

## Discussion

In this study, we provide a ferret dataset of single-cell transcriptomes of neural progenitors covering the entire cortical neurogenesis and early gliogenic phase, revealing their diversity and temporal patterns during cortical development. Comparison of these features between ferrets and humans (*Nowakowski et al., 2017*; *Bhaduri et al., 2021*) indicated at a high resolution that these two gyrencephalic mammals shared a large proportion of progenitor variations and their temporal sequences despite their extremely different neurogenesis timescales. It is remarkable that tRG is conserved

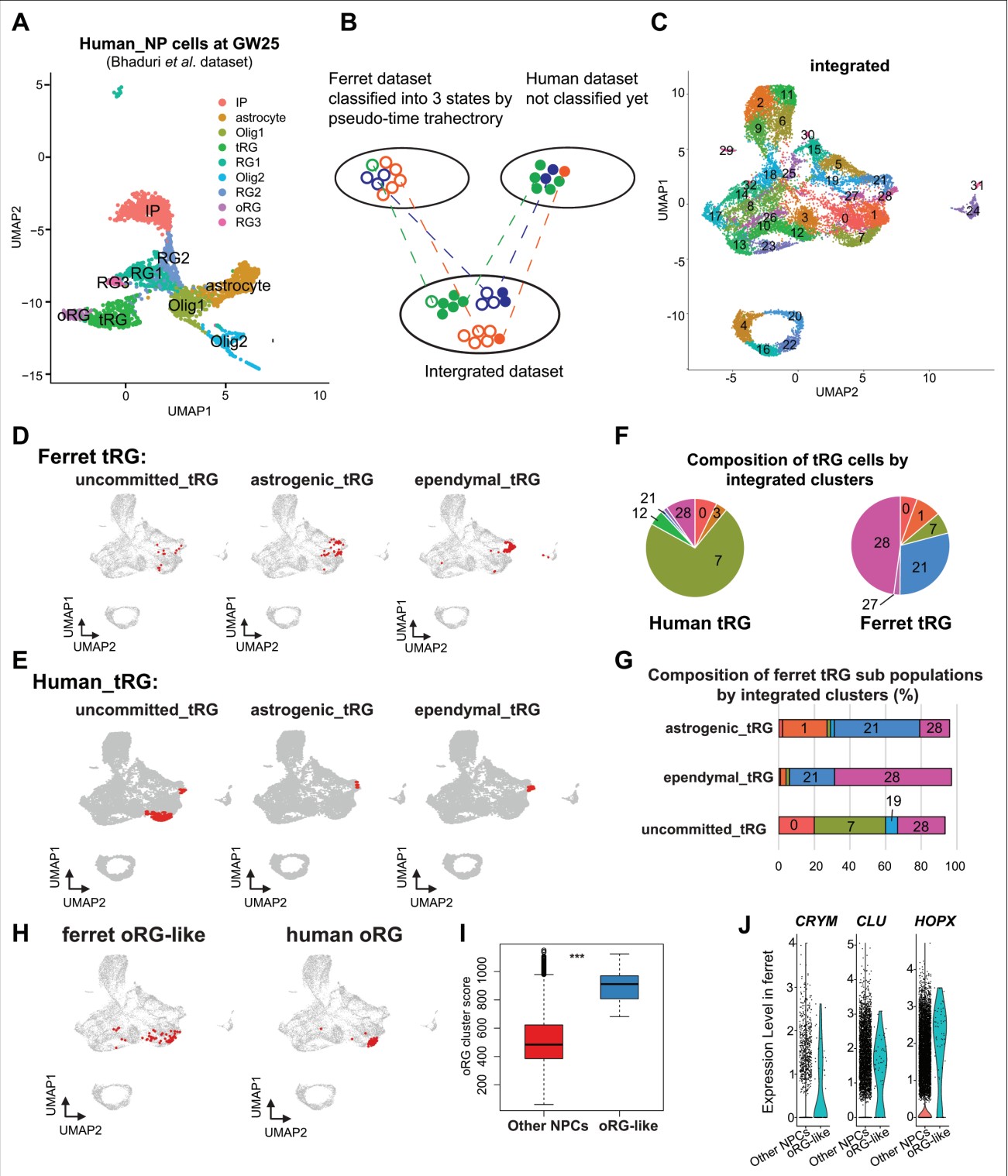

**Figure 7.** Identification of truncated radial glia (tRG) subtypes in humans and outer RG (oRG) in ferrets. (**A**) Uniform Manifold Approximation and Projection (UMAP) visualization of human brain cells at GW25 (and removing neurons and other cell types). Cells are colored by cell type and identified by marker genes (data from *Bhaduri et al., 2021*). (**B**) Schematic of the integration strategy of human and ferret subtypes. We merged the two datasets by pairing mutual nearest neighbor cells (MNN) following canonical correlation analysis (CCA) across species (*Stuart et al., 2019*). (**C**) UMAP visualization of integrated ferret and human datasets colored and numbered by different clusters. (**D, E**) Identification of three tRG subtypes in ferrets (**D**) and humans (**E**). The red dots highlight the indicated tRG subtypes named after pseudo-time trajectory analysis in *Figure 4*: presumptively 'uncommitted', 'astrogenic', and 'ependymal'. (**F**) Distribution of humans (left) and ferrets (right) tRG in the integrated dataset. Human and ferret

*Figure 7 continued on next page*

*Figure 7 continued*

tRG were identified from the separated dataset. The numbering corresponds to cluster numbers in the integrated clustering (**C**) of human and ferret datasets. (**G**) Distribution of ferret tRG subtypes (identified by the pseudo-time analysis) in the integrated dataset. A major cell population of each of the three ferret tRG subtypes (classified by the pseudo-time analysis), more or less, belongs to a single cluster in ferret-human integrated clustering; astrogenic tRG corresponds to cluster 21; ependymal tRG to cluster 20; uncommitted tRG to cluster 7. (**H**) Identification of oRG-like cells in ferret (left) and oRG cells in human (right), by the same way that we have done for tRG. All cells highlighted as red dots. (**I**) oRG scores for oRG-like cells and other neural progenitor cells (NPCs) in ferrets. The result indicates that oRG-like cells share more transcriptomic features than other NPCs, whereas ordinary clustering among ferret NPCs failed to distinguish oRG-like cells from the rest. Box and whisker plots indicate ranges (lines) and upper and lower quartiles with the median (box). (**J**) The expression levels of *CRYM*, *CLU*, and *HOPX* in oRG-like cells and other NPCs in ferrets.

The online version of this article includes the following figure supplement(s) for figure 7:

**Figure supplement 1.** Comparison of the truncated radial glia (tRG) and outer RG (oRG) subtypes between ferrets and humans.

**Figure supplement 2.** The absence of a truncated radial glia (tRG) cluster in human organoid datasets.

between these two phylogenetically distant species. A ferret model allows us to identify the mode of tRG generation, to predict its descendant fates and to analyze cell dynamics in tissues. These findings have not been realized by previous studies, re-evaluating ferrets as a valuable model species to study the neural development of gyrencephalic animals. This comparison between humans and ferrets could not be performed unless genomic information, especially genome-wide gene models, was greatly improved; there had been sufficient genomic information accumulated for ferrets, but mainly for coding regions for genes, which had not been well connected to their 3'-UTR. However, to carry out a large scale of scRNA-seq quantitatively with minimizing the loss of transcript species, accurate 3'-UTR sequences and their link to the coding part are indispensable, because typical scRNA-seq platforms are based on oligo-(dT)-primed cDNA synthesis. We made tremendous efforts to get improved gene models that include 3'-UTR sequences, making it possible for us to compare single-cell transcriptome of ferret cortical progenitors with a huge volume of single-cell transcriptome datasets with high qualities, including those for humans and mice.

## The fate of tRG progenies

Our pseudo-time trajectory analyses and immunohistochemistry analyses suggested that ferret tRG cells differentiate into ependymal cells and astrogenic cells. Although the number tRG cells used in the trajectory analysis was limited (167 cells; *Figure 4*), three tRG subtypes, which are presumably 'precommitted', 'ependymal', and 'astrogenic' tRG, also formed three distinct territories within the original tRG cluster that consisted of 409 cells on UMAP, indicating a consistent heterogeneity within tRG cluster. This mode of the presumptive ependymal cell formation is likely parallel to that of the bifurcation of ependymal cells and adult neural stem cells from the same progenitor population in the striatum on the ventral side (*Ortiz-Álvarez et al., 2019*).

## Comparison of human with ferrets in corticogenesis

Our cross-species comparative analysis predicts that tRG in humans also differentiate into ependymal and astroglial cells although there is no in vivo evidence yet (*Figure 7E*). Furthermore, the rare presence of two presumptively committed tRG subgroups in humans makes this prediction tentative at present. If this prediction is the case, the uncommitted tRG fraction is dominant in GW25 human brains unlike in ferret brains, suggesting that human tRG at GW25 progresses along the differentiation axis less than ferret tRG at P5–P10, most of which are already committed to either ependymal or astrogenic states.

While GW25 has been almost at the latest stages experimentally available from human embryonic brain tissues due to ethical reasons, *Herring et al., 2022* recently reported a large dataset that spans late embryonic to postnatal development in human prefrontal cortex, in which we were not able to detect a tRG population nor other progenitor subtypes, as well as an ependymal cell type, possibly due to regional differences in the collected samples from our ferret dataset. Thus, this postnatal dataset does not seem to be appropriate to compare with our ferret dataset or even human prenatal dataset (*Bhaduri et al., 2020*) to follow tRG progenies (*Figure 7—figure supplement 2*). More enrichment of datasets in the human lateral cortices at the developmental stages after GW25 onward are necessary to test whether our results obtained from ferrets regarding tRG progeny fates can be generalized.

## Commonalities and differences among characteristics of neural progenitors between human ferrets

In this study, our transcriptomic data supported the notion that ferrets are a good model system to investigate mechanisms commonly (at least in both primates and carnivores) underlying the cortical development in gyrencephalic mammals. At the same time, we acknowledge the differences in corticogenesis between ferrets and humans. Most prominently, the timescale of corticogenesis progression is greatly different between the two species, while the variety of NPCs and their temporal patterns of appearances and differentiation exhibit similarities. In addition, the organization of cortical scaffold seems to differ from each other; in human corticogenesis, two major germinal layers, the VZ and OSVZ, become segregated during the expansion phase of RG in the neurogenic stage, due to the loss of vRG, which extend from the ventricular surface to the laminar surface, passing through the OSVZ (*Nowakowski et al., 2016*). In contrast, ferrets maintain vRG with a long radial fiber in the VZ alongside with tRG cells, even beyond the expansion phase of tRG (*Figure 3A*, *Figure 3—figure supplement 1G*).

## Models as human cortical development

Our preliminary analysis of two datasets from human cortical organoids suggests the lack of tRG-like cells and potentially their progeny cell types (*Figure 7—figure supplement 2*, *Bhaduri et al., 2020*; *Herring et al., 2022*). This situation might be improved by optimizing culture protocols. If tRG and ependymal cells, one of the presumptive tRG progeny, are generated in human organoids, as we found in ferrets, it would represent a significant advancement in the human organoid model because ependyma are a fundamental structure for ventricular formation. Further studies using human cortical organoids are expected to be able to generate tRG, and to examine its progeny fates.

In ferrets, genetic manipulations can be achieved through in utero or postnatal electroporation, as well as via virus-mediated transfer of DNA (*Borrell, 2010*; *Kawasaki et al., 2012*; *Matsui et al., 2013*; *Tsunekawa et al., 2016*). Thus, it is theoretically possible to disrupt the *CRYAB* gene in vivo in ferrets to investigate its role in tRG and their progeny, including ependymal cells, and to track the tRG lineage. If the *CRYAB* gene is essential to form ependymal layers, we will be able to explore how the ventricle contributes to cortical folding and expansion. Despite our extensive efforts over a year, we have thus far been unsuccessful in knocking in and/or knocking out the *CRYAB* gene. Nevertheless, we anticipate that technical advances will surpass our expectations, both in ferret and human organoids. Taken together, these functional studies in ferrets as well as in human organoids hold promising insights into the understanding of the tRG lineage and its contribution to cortical development in the near future.

## Materials and methods

### Key resources table

| Reagent type (species) or resource | Designation | Source or reference | Identifiers | Additional information |
|---|---|---|---|---|
| Strain, strain background (*Mustela putorus furo*) | Ferret | Marshall BioResources | N/A | |
| Antibody | Anti-GFP (Chicken polyclonal) | Aves Labs | Cat#GFP-1020; RRID: AB_10000240 | IHC (1:500) |
| Antibody | Anti-Alpha B Crystallin (Mouse monoclonal; clone 1B6.1–3G4) | Abcam | Cat#ab13496; RRID: AB_300400 | IHC (1:500) |
| Antibody | Anti-FoxJ1 (Rabbit monoclonal; clone EPR21874) | Abcam | Cat#ab235445; RRID: N/A | IHC (1:500) |
| Antibody | Anti-Olig2 (Goat polyclonal) | R&D Systems | Cat#AF2418; RRID: AB_2157554 | IHC (1:500) |
| Antibody | Anti-EOMES (Rat monoclonal; clone Dan11mag, eFluor 660, eBioscience) | Thermo Fisher Scientific | Cat#50-4875-82; RRID: AB_2574227 | IHC (1:500) |
| Antibody | Anti-Pax6 (Rabbit polyclonal) | Covance | PRB-278P; RRID: AB_291612 | IHC (1:500) |

*Continued on next page*

*Continued*

| Reagent type (species) or resource | Designation | Source or reference | Identifiers | Additional information |
|---|---|---|---|---|
| Antibody | Anti-GFAP (mouse) | Sigma-Aldrich | Cat# G3893, RRID: AB_477010 | IHC (1:500) |
| Antibody | Anti-HOPX (Rabbit polyclonal; clone FL-73) | Santa Cruz Biotechnology | Cat#sc-30216; RRID: AB_2120833 | IHC (1:500) |
| Antibody | Anti-RFP (Rat monoclonal; clone 5F8) | ChromoTek | Cat#5f8-100; RRID: AB_2336064 | IHC (1:500) |
| Antibody | Anti-A cyclase III (Goat polyclonal; clone N-14) | Santa Cruz Biotechnology | Cat#sc-32113; RRID: AB_2223118 | IHC (1:250) |
| Antibody | Anti-Ctip2 (Rat monoclonal; clone 25B6) | Abcam | Cat#ab18465; RRID: AB_2064130 | IHC (1:500) |
| Antibody | Anti-SATB2 (Mouse monoclonal; clone SATBA4B10) | Abcam | Cat#ab51502; RRID: AB_882455 | IHC (1:500) |
| Antibody | Ki67 (Rat monoclonal; clone SolA15, eFluor 660, eBioscience) | Thermo Fisher Scientific | Cat#50-5698-82; RRID: AB_257423 | IHC (1:500) |
| Antibody | Anti-Mouse IgG AF488 (Donkey) | Jackson ImmunoResearch Labs | Cat#715-545-151; RRID: AB_2341099 | IHC (1:500) |
| Antibody | Donkey anti-Mouse IgG Cy3 | Jackson ImmunoResearch Labs | Cat#715-165-151; RRID: AB_2315777 | IHC (1:500) |
| Antibody | Anti-Mouse IgG 647 (Donkey) | Jackson ImmunoResearch Labs | Cat:715-605-151; RRID: AB_2340863 | IHC (1:500) |
| Antibody | Anti-Rat IgG 647 (Donkey) | Jackson ImmunoResearch Labs | Cat#712-605-153; RRID: AB_2340694 | IHC (1:500) |
| Antibody | Anti-Rat IgG Cy3 (Donkey) | Jackson ImmunoResearch Labs | Cat#712-165-153; RRID: AB_2340667 | IHC (1:500) |
| Antibody | Anti-Rabbit IgG Cy3 (Donkey) | Jackson ImmunoResearch Labs | Cat#711-165-152; RRID: AB_2307443 | IHC (1:500) |
| Antibody | Anti-Rabbit IgG AF647 (Donkey) | Jackson ImmunoResearch Labs | Cat#711-605-152; RRID: AB_2492288 | IHC (1:500) |
| Antibody | Anti-Goat IgG AF488 (Donkey) | Jackson ImmunoResearch Labs | Cat#705-545-003; RRID: AB_2340428 | IHC (1:500) |
| Antibody | Anti-Goat IgG AF647 (Donkey) | Jackson ImmunoResearch Labs | Cat#705-605-147; RRID: AB_2340437 | IHC (1:500) |
| Antibody | Anti-Chicken IgG Alexa Fluor 488 (Donkey polyclonal) | Jackson ImmunoResearch Labs | Cat#703-545-155; RRID: AB_2340375 | IHC (1:500) |
| Recombinant DNA reagent | phmAG1-S1 (humanized monomeric Azami Green 1) | MBL | Cat#AM-V0034 | |
| Recombinant DNA reagent | Plasmid: pLR5-Hes5-d2-Azamigreen | This paper | N/A | |
| Recombinant DNA reagent | Plasmid: pPB-LR5-mCherry | This paper | N/A | |
| Recombinant DNA reagent | Plasmid: pCAX-hyPBase | *Fujita et al., 2020* | N/A | |
| Recombinant DNA reagent | Plasmid: pCAG-Cre | *Fujita et al., 2020* | N/A | |
| Recombinant DNA reagent | Plasmid: pBP-LR5-floxstop-EGFP | *Fujita et al., 2020* | N/A | |
| Recombinant DNA reagent | Plasmid: pCR-BluntII-TOPO-Clu-probe | This paper | N/A | |
| Chemical compound, drug | Dulbecco's Modified Eagle Medium (DMEM) | Nacalai Tesque | Cat#08458-45 | |

*Continued on next page*

*Continued*

| Reagent type (species) or resource | Designation | Source or reference | Identifiers | Additional information |
|---|---|---|---|---|
| Chemical compound, drug | Dulbecco's Modified Eagle Medium (DMEM) F12+GlutaMax | Gibco | Cat#10565-018 | |
| Chemical compound, drug | BSA Fraction V (7.5%) | Gibco | Cat#15260-037 | |
| Chemical compound, drug | EDTA (0.5M) | Invitrogen | Cat#AM9260G | |
| Chemical compound, drug | Papain | Funakoshi | Cat#WOR-1231-78 | |
| Chemical compound, drug | Hank's Balanced Salt Solution, HBSS (-) | Nacalai Tesque | Cat#17460-15 | |
| Chemical compound, drug | Fast Green | Wako Pure Chemical Industries | Cat#2353-45-9 | |
| Chemical compound, drug | Human FGF-basic | Peprotech | Cat#100-18B-10UG | |
| Chemical compound, drug | B27 Supplement (50x) | Gibco | Cat#12587-010 | |
| Chemical compound, drug | Penicillin/Streptomycin | Nacalai Tesque | Cat#09367-34 | |
| Chemical compound, drug | HistoVT one | Nacalai Tesque | Cat#06380-05 | |
| Chemical compound, drug | TissueTek O.C.T. compound | Sakura | Cat#4583 | |
| Chemical compound, drug | Triton X-100 | Bio-Rad Laboratories | Cat#1610407 | |
| Chemical compound, drug | Tet System Approved FBS US-sourced | Clontech | Cat#631105 | |
| Chemical compound, drug | Donkey serum | Sigma | Cat#S30 | |
| Chemical compound, drug | DAPI | Nacalai Tesque | Cat#1034-56 | |
| Chemical compound, drug | Mountant (PermaFluor) | Thermo Fisher Scientific | TA-006-FM | |
| Chemical compound, drug | UltraPure Low Melting Point agarose | Thermo Fisher Scientific | Cat#16520050 | |
| Chemical compound, drug | Aqueous Mounting Medium PermaFluor | Thermo Fisher Scientific | Cat#TA-030-FM | |
| Chemical compound, drug | DIG RNA labeling Mix, 10x | Roche | Cat#11277073910 | |
| Chemical compound, drug | Anti-DIG-AP Fab fragments | Roche | Cat#11093274910 | |
| Chemical compound, drug | 20x SSC (pH 4.5) | Invitrogen | Cat#AM9763 | |
| Chemical compound, drug | Formamide | Wako | Cat#11-0740-5 | |
| Chemical compound, drug | Formaldehyde solution | Sigma | Cat#11-0720-5 | |
| Chemical compound, drug | Brewer's yeast tRNA | Roche | Cat#10109525001 | |
| Chemical compound, drug | Acetylated BSA | Nacalai Tesque | Cat#01278-44 | |

*Continued on next page*

*Continued*

| Reagent type (species) or resource | Designation | Source or reference | Identifiers | Additional information |
|---|---|---|---|---|
| Chemical compound, drug | Heparin | Sigma | Cat#9041-08-1 | |
| Chemical compound, drug | NBT/BCIP Stock solution | Roche | Cat#11681451001 | |
| Chemical compound, drug | Proteinase K | Roche | Cat#03115887001 | |
| Chemical compound, drug | Proteinase K | QIAGEN | Cat#158920 | |
| Chemical compound, drug | RNase A | QIAGEN | Cat#158924 | |
| Chemical compound, drug | Agarase | Life Technologies | Cat#EO0461 | |
| Chemical compound, drug | T7 RNA polymerase | Roche | Car#10881767001 | |
| Sequence-based reagent | CLU_F | This paper | PCR primers | GAATGACACCAAGGATTCAGAAACGAAGCT |
| Sequence-based reagent | CLU_R | This paper | PCR primers | ATGGAATTCACAGAAGAAGACAACCAGGAC |
| Commercial assay or kit | Chromium Single Cell 3′ Library & Gel Bead Kit v2 | 10x Genomics | Cat#120267 | |
| Commercial assay or kit | Chromium i7 Multiplex Kit Kit | 10x Genomics | Cat#120262 | |
| Commercial assay or kit | Chromium Single Cell A Chip | 10x Genomics | Cat#1000009 | |
| Commercial assay or kit | Chromium Controller & Accessory Kit | 10x Genomics | Cat#120223 | |
| Commercial assay or kit | Chromium Genome Reagent Kit v1 Chemistry | 10x Genomics | Cat#PN-120216 | |
| Commercial assay or kit | HiSeq PE Rapid Cluster Kit v2 | Illumina | Cat#PE-402-4002 | |
| Commercial assay or kit | TruSeq PE Cluster Kit v3-cBot-HS | Illumina | Cat#PE-401-3001 | |
| Commercial assay or kit | Qubit RNA HS Assay Kit | Thermo Fisher Scientific | Cat#Q32852 | |
| Commercial assay or kit | RNA 6000 Nano kit | Agilent Technologies | Cat#5067-1511 | |
| Commercial assay or kit | RNA 6000 Nano kit | Agilent Technologies | Cat#5067-1511 | |
| Commercial assay or kit | TruSeq Stranded mRNA Sample Prep Kit | Illumina | Cat#20020594 | |
| Commercial assay or kit | TruSeq single-index adaptor kit | Illumina | Cat#20015960 | |
| Commercial assay or kit | KAPA Real-Time Library Amplification Kit | Roche | Cat#KK2611 | |
| Commercial assay or kit | PrimeScript II 1st strand cDNA Synthesis Kit | Takara | Cat#6210A | |
| Commercial assay or kit | PrimeScript 1st strand cDNA Synthesis Kit | Takara | Cat#6110A | |
| Commercial assay or kit | RNeasy mini kit | QIAGEN | Cat#74106 | |

*Continued on next page*

*Continued*

| Reagent type (species) or resource | Designation | Source or reference | Identifiers | Additional information |
|---|---|---|---|---|
| Commercial assay or kit | CHEF Mammalian Genomic DNA Plug Kit | Bio-Rad Laboratories | Cat#1703591 | |
| Commercial assay or kit | Zero Blunt TOPO PCR Cloning Kit | Thermo Fisher Scientific | Cat#450245 | |
| Software, algorithm | R (v.3.6.3 2020-02-29) | R project for Statistical Computing | https://www.r-project.org/; RRID: SCR_001905 | |
| Software, algorithm | R version 4.1.2 | R project for Statistical Computing | https://www.r-project.org/; RRID: SCR_001905 | |
| Software, algorithm | Seurat v3 | *Stuart et al., 2019* | RRID: SCR_016341; https://satijalab.org/seurat/get_started.html | |
| Software, algorithm | Monocle2 | *Qiu et al., 2017*; *Trapnell et al., 2014* | RRID: SCR_016339; http://cole-trapnell-lab.github.io/monocle-release/docs/ | |
| Software, algorithm | Enrichr | *Chen et al., 2013*; *Kuleshov et al., 2016* | http://amp.pharm.mssm.edu/Enrichr/; RRID: SCR_001575 | |
| Software, algorithm | Cell Ranger v2 | 10x Genomics | RRID: SCR_017344; https://support.10xgenomics.com/single-cell-gene-expression/software/pipelines/latest/what-is-cell-ranger | |
| Software, algorithm | ggplot2 | *Wickham, 2016* | https://ggplot2.tidyverse.org | |
| Software, algorithm | pheatmap | N/A | https://rdrr.io/cran/pheatmap/ | |
| Software, algorithm | bcl2fastq ver. 1.8.4 | Illumina | RRID:SCR_015058; https://support.illumina.com/sequencing/sequencing_software/bcl2fastq-conversion-software.html | |
| Software, algorithm | Real Time Analysis ver. 1.18.64 | Illumina | RRID:SCR_014332; http://support.illumina.com/sequencing/sequencing_software/real-time_analysis_rta.html | |
| Software, algorithm | Supernova ver. 2.0.0 | 10x Genomics | https://support.10xgenomics.com/de-novo-assembly/software/overview/latest/welcome | |
| Software, algorithm | RepeatMasker ver. 4.0.7 | *Smit et al., 2013* | RRID:SCR_012954; http://repeatmasker.org/ | |
| Software, algorithm | Repbase ver. 23.01 | N/A | RRID:SCR_021169; https://www.girinst.org/repbase/ | |
| Software, algorithm | BRAKER ver. 2.0.5 | *Hoff et al., 2019* | RRID:SCR_018964; https://github.com/Gaius-Augustus/BRAKER; *Stanke et al., 2023* | |
| Software, algorithm | GeneMark-ET ver. 4.33 | *Lomsadze et al., 2014* | http://topaz.gatech.edu/GeneMark/ | |
| Software, algorithm | AUGUSTUS ver. 3.3 | *Stanke et al., 2008* | RRID:SCR_008417; http://bioinf.uni-greifswald.de/augustus/ | |
| Software, algorithm | STAR | *Dobin et al., 2013* | https://github.com/alexdobin/STAR (*Dobin, 2023*) | |
| Software, algorithm | HISAT2 ver. 2.1.0 | *Kim et al., 2019* | RRID:SCR_015530; http://ccb.jhu.edu/software/hisat2/index.shtml | |
| Software, algorithm | StringTie ver. 1.3.4d | *Kovaka et al., 2019* | RRID:SCR_016323; https://ccb.jhu.edu/software/stringtie/ | |

*Continued on next page*

*Continued*

| Reagent type (species) or resource | Designation | Source or reference | Identifiers | Additional information |
|---|---|---|---|---|
| Software, algorithm | GMAP ver. 2017-11-15 | *Wu and Watanabe, 2005* | RRID:SCR_008992 | |
| Software, algorithm | gVolante ver. 1.2.1 | *Nishimura et al., 2017* | https://gvolante.riken.jp | |
| Software, algorithm | Velocity 3D Image Analysis Software | Perkin Elmer | RRID: SCR_002668 | |
| Software, algorithm | MetaMorph | Molecular Devices | https://www.moleculardevices.com/products/cellular-imaging-systems/acquisition-and-analysis-software/metamorph-microscopy#gref | |
| Software, algorithm | Adobe Illustrator | Adobe | RRID: SCR_010279; http://www.adobe.com/products/illustrator.html | |
| Software, algorithm | ImageJ (Fiji) | *Schindelin et al., 2012* | RRID: SCR_002285; http://fiji.sc | |
| Other | CUBIC solution 2 | *Susaki et al., 2015* | N/A | |
| Other | SH800 Cell Sorter | SONY | https://www.sonybiotechnology.com | |
| Other | Countess Cell Counter | Thermo Fisher Scientific | https://www.thermofisher.com | |
| Other | Countess II Cell Counter | Thermo Fisher Scientific | https://www.thermofisher.com | |
| Other | Cryostat CM3050 S | Leica | https://www.leicabiosystems.com | |
| Other | Liner Slicer vibratome | Dosaka EM | http://www.dosaka-em.jp | |
| Other | FV1000 confocal microscope | Olympus | https://www.olympus-lifescience.com | |
| Other | scRNA-seq data from developing human cerebral cortex | *Nowakowski et al., 2017* | https://cells.ucsc.edu/?ds=cortex-dev | |
| Other | scRNA-seq data from GW25 human cerebral cortex | *Bhaduri et al., 2021* | https://kriegsteinlab.ucsf.edu/datasets/arealization | |
| Other | scRNA-seq data from human organoids | *Herring et al., 2022* | http://development.psychencode.org/ | |
| Other | scRNA-seq data from human primary brain tissue and organoids | *Bhaduri et al., 2020* | https://organoidreportcard.cells.ucsc.edu | |

## Ferret brain tissue samples

Pregnant ferrets were purchased from Marshall BioResources. We also bred them to maintain a population that is sufficient to regularly get pregnant females in the animal facility of RIKEN Center for Biosystems Dynamics Research under the license given from Marshall BioResources. All procedures during animal experiments were performed in accordance with the legal and institutional ethical regulations of RIKEN Center for Biosystems Dynamics Research.

Ferret pups are born at day 41 or day 42 of gestation. The day of birth was counted as postnatal day 0 (P0). Ferret embryos and pups were euthanized prior to brain removal.

Samples used for scRNA-seq were obtained from one or two pups for each developmental stage: embryonic days 25, 34, and 40 and postnatal days 1, 5, and 10. Samples used for immunohistology-based quantifications were obtained from two or three pups from different mothers. Samples used for a time-lapse imaging were obtained from one animal per experiment, which is performed four times, independently.

## Gene model construction of the ferret genome

In this study, we used a ferret genome reference with expanded reference gene annotations with DDBJ (detail see below). Briefly, gene models were constructed using Chromium by tagging all fragments from a long genomic DNA in a droplet, so that sequences from long genomic DNAs could be

successfully aligned to cover so far unconnected genomic regions in the NCBI (NCBI Assembly ID: 286418 (MusPutFur1.0)) database. Detailed information after mapping is listed in *Supplementary file 1A and B*.

## 1) RNA-seq library preparation and sequencing

Total RNA was extracted from the embryonic tissues (*Supplementary file 7*) using the RNeasy mini kit (QIAGEN). Concentration and length distribution of the RNA were checked with the Qubit RNA HS Assay Kit on a Qubit 2.0 Fluorometer (Thermo Fisher Scientific) and the RNA 6000 Nano Kit on a 2100 Bioanalyzer (Agilent Technologies). Sequencing libraries were prepared using 1μg of total RNA with the TruSeq Stranded mRNA Sample Prep Kit (Illumina) and the TruSeq single-index adaptor (Illumina), except for the sample 'gonad and mesonephros' (see *Supplementary file 7*) for which 500ng of total RNA was used. The RNA was fragmented at 94°C for either 8min or 2min, which resulted in libraries with variable insert size distributions (*Hara et al., 2015*). The optimal numbers of PCR cycles were pre-determined using the KAPA Real-Time Library Amplification Kit (Roche) as described previously (*Tane-gashima et al., 2018*). Libraries were sequenced with the Rapid Run mode of HiSeq1500 (Illumina) using the HiSeq PE Rapid Cluster Kit ver. 2 (Illumina) and the HiSeq Rapid SBS Kit v2-HS (Illumina) to produce paired-end reads of 127 nucleotides. Base calling was performed with RTA ver. 1.18.64, and the fastq files were generated with bcl2fastq ver. 1.8.4 (Illumina).

## 2) Genomic DNA extraction and chromium genome library construction

Genomic DNA was extracted from a liver of an adult female ferret using the CHEF Mammalian Genomic DNA Plug Kit (Bio-Rad, Cat. No. #1703591). In brief, the liver tissue of about 50mg was homogenized in PBS (-) with a Dounce tissue grinder on ice, fixed with ethanol, and subsequently embedded in low melting point agarose gel (Bio-Rad). After the protein digestion with Proteinase K (QIAGEN, Cat. No. #158920) and RNA digestion with RNase A (QIAGEN, Cat. No. #158924), the gel plugs were further digested by Agarase (Life Technologies, Cat. No. #EO0461). DNA was purified by the drop dialysis method using the MF-Millipore Membrane Filter (Merck Millipore, Cat. No. #VCWP04700). Length distribution of the genomic DNA was analyzed by pulsed-field gel electrophoresis, which exhibited an average length of over 2 Mbp. The Chromium library was constructed using the Chromium Genome Reagent Kit v1 Chemistry (10x Genomics, Cat No. #PN-120216), and sequencing was performed with an Illumina HiSeq X to obtain paired-end 151-nt-long reads.

## 3) Genome assembly

De novo genome assembly using the Chromium linked reads was performed with Supernova ver. 2.0.0 (10x Genomics) with the default parameters, and the resultant sequences were output with the option 'pseudohap'. Detection and masking of repetitive sequences were performed using RepeatMasker ver. 4.0.7 (*Smit et al., 2013*) with NCBI RMBlast ver. 2.6.0+ andthe species-specific repeat library from RepBase ver. 23.01.

## 4) Gene model construction

Gene models for gene expression level quantification were constructed in the three following steps. First, ab initio gene prediction was performed by BRAKER ver. 2.0.5 (*Hoff et al., 2019*). In this process, GeneMark-ET ver. 4.33 (*Lomsadze et al., 2014*) with the information of spliced RNA-seq alignment was used for training, and AUGUSTUS ver. 3.3 (*Stanke et al., 2008*) with the parameters of 'UTR = on, species = human' was used for gene prediction. For the abovementioned training of BRAKER, the RNA-seq reads were aligned to the repeat-masked genome sequences by STAR ver. 2.5.4a (*Dobin et al., 2013*) with default parameters using the entire set of RNA-seq reads of various tissue types (*Supplementary file 7*). Second, the information of RNA-seq alignment was directly incorporated into the gene models to improve the coverage of the 3'-UTR region of each gene which is not reliably predicted in ab initio gene prediction. This computation was performed with HISAT2 ver. 2.1.0 (*Kim et al., 2019*) and StringTie ver. 1.3.4d (*Kovaka et al., 2019*), whose output GTF file was merged with the gene models produced by BRAKER using the merge function of StringTie. Finally, the existing transcript sequences of MusPutFur1.0 available at NCBI RefSeq (GCF_000215625.1) was incorporated into the gene models by mapping their sequences to the repeat-masked genome sequences using

GMAP ver. 2017-11-15 (*Wu and Watanabe, 2005*) with the 'not report chimeric alignments' option. The gene name of each locus was adopted from the annotation in RefSeq, and the gene name of a newly predicted locus was assigned according to the results of a BLASTX search against the UniProtKB Swiss-Prot database release 2020_01. The assignment was performed only for genes with a bit score (in the abovementioned BLASTX search) of greater than 60. When a locus has multiple transcripts, the one with the highest score was adopted for gene naming. If a transcript estimated by ab initio prediction bridged multiple genes, they were incorporated into the gene models as separate genes.

### 5) Completeness assessment of assemblies

To assess the continuity of the genome assembly and gene space completeness of the gene models, gVolante ver. 1.2.1 (*Nishimura et al., 2017*) was used with the CEGMA ortholog search pipeline and the reference orthologs gene set CVG (*Hara et al., 2015*).

## In utero and postnatal electroporation

IUE and postnatal electroporation in ferrets was performed as described previously (*Kawasaki et al., 2012*; *Matsui et al., 2013*; *Tsunekawa et al., 2016* for IUE; *Borrell, 2010* for postnatal electroporation) with modifications. Briefly, pregnant ferrets or ferret pups were anesthetized with isoflurane at indicated stages of the development. The location of lateral ventricles was visualized with transmitted light delivered through an optical fiber cable. Three μl of plasmid DNA solution was injected into the lateral ventricle at indicated developmental stages using an injector. Each embryo or pup was placed between the paddles of electrodes, and was applied a voltage pulse of 45V at the duration from 100 ms to 900ms (100,0 Pon and 1000,0 Poff) 10 times for IUE, and under the same condition except a voltage of 60V for postnatal electroporation (CUY21 electroporator, Nepa Gene).

## Plasmids

To assure a stabilized expression of transgenes, we combined expression vectors with a hyperactive piggyBac transposase system to integrate expression vectors into the genome for a stable expression (*Yusa et al., 2011*). To enrich NPC populations in 'AG' samples used for the single-cell transcriptome analysis, pLR5-Hes5-d2-AzamiGreen (0.5μg/μl; Hes5 promoter was gifted from Kageyama Laboratory; *Ohtsuka et al., 2006*) was electroporated at E30 for E40, or at E34 for postnatal samples, together with pCAX-hyPBase (0.5μg/μl). Among them, for P1 and P10 samples, pPB-LR5-mCherry (0.5μg/μl) was also included in the DNA solution.

To sparsely label the cell cytoplasm for detailed imaging of individual cells on vibratome-cut thick cortical sections, we have used pPB-LR5-floxstop-EGFP combined with a low concentration of Cre-expressing plasmid. The concentrations used for these plasmids were 0.5μg/μl or 1.0μg/μl for a GFP labeling and 1ng/μl or 10ng/μl of Cre-expressing plasmid for P0 or P5 samples, respectively (*Figure 3A*). For all electroporation experiments, Fast Green solution (0.1mg/ml; Wako Pure Chemical Industries) was added into the freshly prepared mixture of plasmid DNA to visualize the injection.

For in situ hybridization probes, the PCR product was inserted into pCR-BluntII-TOPO for cloning and sequencing.

## scRNA-seq libraries

10x v2 Chromium was performed on dissociated single cells according to the manufacturer's protocol.

### 1) Single-cell isolation and sorting of AG-expressing cells

Cell suspension was prepared as reported previously (*Wu et al., 2022*) with modifications. Brains were collected at indicated stages, transferred in the ice-cold dissection Dulbecco's Modified Eagle Medium (DMEM) F12, and meninges were removed. Somatosensory area was dissected from embryonic or postnatal cortices using an ophthalmic knife (15°) under a dissection microscope. Sliced tissues were dissociated via enzymatic digestion with papain at 37°C for 30–45min in ice-cold Hank's Balanced Salt Solution (HBSS) (-) with EDTA (0.1M). Dissociated cells were centrifuged at 1000 × *g* for 5min to remove papain by washing with PBS and were resuspended in 0.375% BSA/HBSS (-), or in the sorting buffer (DMEM F12+ GlutamaX [Thermo Fisher]; 0.1% of Penicillin/Streptomycin [Millipore]; 20ng/ml human basic FGF, Peprotech; 1xB27 RA-, Gibco) for cell sorting (see below). Homogenous cell suspension filtered through 35μm strainer (Falcon).

To concentrate NPC populations at the expense of mature neurons, two methods were used for each sample, except for E25, from which cells were collected using whole somatosensory area:

1. 'T' samples: the CP of cortical sections were dissected out and a part of the IZ was likely to be included in the discarded region.
2. 'AG' samples: brains collected at indicated stages were electroporated in utero by AG expression vector under the control of Hes5 promoter. The latter method allows the expression of AG in only NPC by the presence of a degradation signal 'd2' in the vector to degrade AG protein in HES5-negative differentiating progeny. Only 'AG' samples were processed with cell sorting and dissociated cell obtained as described above were placed in an SH800 cell sorter (SONY) to sort AG-positive cells in 0.375%BSA/HBSS(-) solution. Cell survival and cell number were quantified by Countess or Countess II (Invitrogen), prior to an application of single-cell isolation using 10x v2 Chromium kit.

Samples from the same developmental stage, except for P10 'T' sample, were born from the same mother and were collected by applying either of above methods on the same day and processed in parallel.

### 2) Library preparation and sequencing

Single-cell libraries were generated according to the manufacturer's instructions. Cell suspensions were diluted for an appropriate concentration to obtain 3000cells per channel of a 10x microfluidic chip device and were then loaded on the 10x Chromium chips accordingly to the manufacturer's instructions.

Total cDNA integrity and quality were assessed with Agilent 2100 Bioanalyzer.

Libraries were sequenced on the HiSeq PE Rapid Cluster Kit v2 (Illumina), or the TruSeq PE Cluster Kit v3-cBot-HS, to obtain paired-end 26 nt (Read 1) to 98 nt (Read 2) reads.

## Immunohistology and confocal imaging

Ferret brains were removed from embryos or pups and fixed for one or two overnights, respectively, in 1% paraformaldehyde (PFA) prepared in 0.1M phosphate buffer (PB, pH 7.4) at 4°C. P35 ferrets were transcardially perfused with cold PBS, followed by 4% PFA, under deep anesthesia with isoflurane, then collected brains were post-fixed with 1% PFA. After the fixation of brains or cortical tissue slices after a live imaging, they were equilibrated in 25% sucrose overnight before embedding in O.C.T. compound (TissueTek, Sakura) and frozen in liquid nitrogen. Frozen samples were stored at –80°C prior to a coronal sectioning using a cryostat (CM3050S Leica Microsystems, 12μm sections). After equilibration at room temperature (RT), sections were washed in PBS with 0.1% Tween (PBST), followed by a treatment with an antigen retrieval solution of HistoVT one (Nacalai Tesque) diluted 10 times in milliQ water, at 70°C for 20min. Sections were then blocked 1hr at the RT in PBS with 2% Triton-X100 and 2% normal donkey serum (Sigma), followed by an incubation overnight at 4°C with primary antibody diluted in the blocking solution. After washing in PBST three times, sections were treated with appropriate fluorescence-conjugated secondary antibodies (1:500) along with 4',6-diamidino-2-phenylindole (DAPI, 1:1000) for 1hr at RT. Sections were washed again prior to mounting with PermaFluor solution (Thermo Fisher Scientific).

Immunostaining of thick ferret brain sections (200μm) were performed as previously reported (*Tsunekawa et al., 2016*). Briefly, ferret brains were fixed in 1% PFA, washed in PB overnight at 4°C, and embedded in 4% low-melting agarose (UltraPure LMP agarose, Thermo Fisher Scientific). Embedded brains were sliced coronally at 200μm thickness by a vibratome (LinerSlicer, DOSAKA EM) on ice. Floating sections were washed three times with PBST, treated with the blocking solution for 1hr at RT, and incubated with primary antibodies for five or six overnights at 4°C under shaking. Sections were then washed in PBST and treated with secondary antibodies for five or six overnights at 4°C under shaking. After washing, we mounted brain slices with CUBIC solution 2 to allow the transparency.

Fluorescent images were acquired using an FV1000 confocal microscope (Olympus, Japan). Somatosensory cortices were captured with 20× or 100× objective lenses. When 100× objective lens was used, z-stacks of confocal images were taken with an optical slice thickness 1.0μm or 1.5μm and MAX-projection images obtained by FiJi are shown.

All the antibodies and reagents used in this study are listed in Key resources table.

## Probe preparation

Primers for PCR targeting ferret *CLU* gene were designed with Primer3 ver 0.4.0 software. Total RNA was isolated from embryonic ferret brain, collected in Trizol. cDNA was generated from total RNA using the Prime Script 1st strand cDNA synthesis kit (Takara) according to the manufacturer's recommended procedure. Target cDNA was inserted into pCR-TOPOII-Blunt plasmid for cloning and sequencing using M13F or M13R primers. Anti-sense cRNA probes were then generated by in vitro transcription using T7 promoter.

## In situ hybridization

ISH was performed as described previously, with some modifications (*Mashiko et al., 2012*). Briefly, ferret brains were perfused in 4% PFA in PBS and were fixed at 4% PFA in 0.1M PB. Fixed brain were incubated in 30% sucrose/4% PFA in 0.1M PB for at least 1day and were stored at –80°C until ISH. Frozen brains were sectioned in coronal plane at 25µm using cryostat (CM3050S Leica Microsystems). Sections on slides were post-fixed at 4% PFA in 0.1M PB and treated with proteinase K (Roche). Sections were hybridized with digoxigenin (DIG)-labeled probes at 72°C overnight in hybridization solution. Sections were then washed and blocked with donkey serum and incubated with pre-absorbed DIG antibody, conjugated to alkaline phosphatase for 2hr at RT. Color development was performed in solution containing NBT/BCIP (Roche).

## Quantification and statistical analysis

Statistical details including experimental n, statistical tests, and significance are reported in Figure Legends.

## Transcriptome analyses

### 1) Alignment and raw processing of data

Fastq files were obtained from individual samples and were processed using Cell Ranger pipeline v2. Alignment was done using 'Cell Ranger count' function with default parameters accordingly to the manufacturer's instructions to map reads to the ferret reference (MPF_Kobe 2.0.27). The raw data for each set of cells within a sample was obtained by cellranger count function and was read using Seurat 'Read10X' function (Seurat v3.1.5) (*Stuart et al., 2019*), creating a matrix for unique molecular identified (UMI) counts of each gene within each cell.

### 2) Filtering and normalization

Filtering and normalization were performed using Seurat (v3.1.5) (*Stuart et al., 2019*). Briefly, each sample was filtered by removing low-quality cells with unique feature counts less than 200, and genes expressed in less than threecells (Seurat function 'CreateSeuratObject'). Different samples were then merged using the Seurat 'merge' function. Merged data was further subset by keeping cells with features (genes) over 200 and less than 5000. Raw UMI counts were then normalized using 'LogNormalize' as normalization method, which divided reads by the total number of UMIs per cell, then multiplied by 10,000 (Seurat 'NormalizeData' function). This resulted in a total of 30,234cells and 19,492 genes: E25 (3486cells), E34AG (3223cells), E34T (2260cells), E40AG (1102), E40T (1581cells), P1AG (2681), P1T (3641), P5AG (2429), P5T (2926), P10AG (3010), and P10T (3895).

### 3) Single-cell clustering and visualization

Cell clustering was employed to the entire cell population after removing low-quality cells, using Seurat (v3.1.5) (*Stuart et al., 2019*). 2000 highly variable genes were identified and used for the downstream analysis using Seurat 'FindVariableFeatures' function (selection.method = 'vst'), which allowed the calculation of average expression and dispersion for each gene. Normalized data was then processed for scaling using Seurat function 'Scale Data' with default settings, which also allowed the regression of the batch. Principal component analysis (PCA) was performed with 2000 variable genes to reduce dimensionality of the scaled dataset and 50 PCs were retained (Seurat 'RunPCA' function). Clustering was then performed using graph-based clustering approach ('Find Neighbors' function) using top 20 PCs, which were selected based on the standard deviation of PCs on the elbowplot created by Seurat function 'ElbowPlot' and based on statistical significance calculated by JackStraw application. Briefly,

cells are embedded in a k-nearest neighbor (KNN) graph based on the Euclidean distance in a PCA space. Then, this KNN graph is used to group cells on a shared nearest neighbor graph based on calculations of overlap between cells with similar gene expression patterns (Jaccard similarity). Cells were then clustered by Louvain algorithm implemented in Seurat 'ClusterCells' function (resolution = 0.8, dims = 1:20). Next, we used Seurat 'RunUMAP' function, which resulted in cell clusters being separated in embedding space while preserving the balance between local and global structure.

## 4) Cluster annotations

Cluster markers were obtained using 'FindAllMarkers' Seurat function. We tested genes that showed at least a 0.25-fold difference between the cells in the cluster and all remaining cells, and that were detected in more than 25% of the cells in the cluster. Clusters, which we called subtypes, were then annotated by comparing cluster markers to previously identified cell type markers in the literature for mouse and human datasets. The full list of markers is given under *Supplementary file 1D*. The heatmap in *Figure 2B* shows the expression data for the top 10 highly expressed marker genes for each cluster using a downsampling of maximum 500cells from each cluster. Plotting cells onto the UMAP plot by their batch indicated that batches did not influence clustering (*Figure 2—figure supplement 1B*) in accordance with the differential expression of cluster markers (*Figure 2B*, *Figure 2—figure supplement 2A*). When a cluster was enriched based on the developmental stage, the information was included in the annotation. For example, 'early RG' cluster was enriched in E25 samples, and expressed both common RG cell markers shared with other RG clusters in our dataset, but also expressed early onset genes reported in mouse studies (*Okamoto et al., 2016*; *Telley et al., 2019*), such as *HMGA2*, *FLRT3*, *LRRN1* (*Supplementary file 1D*). Therefore, we named this cluster as 'early RG' based on its age-dependent properties.

To achieve more precise clustering, cycling cells were identified with known markers implemented in Seurat package (S genes and G2M genes). Clusters enriched in S genes or G2M genes were identified, including early_RG2 (S), early_RG3 (G2M), late_RG2 (G2M), late_RG3 (S), IPC2 (S), and IPC3 (G2M) (*Figure 2—figure supplement 2C*). Subtypes of RG and IPC clusters were further combined to facilitate the data representation (*Figure 2B*).

## 5) Differential gene expression analysis

Differential gene expression analysis was assessed using 'FindMarkers' Seurat function with default parameters.

## 6) Pseudo-time analysis

Monocle 2 was used to construct developmental trajectories based on pseudo-time ordering of single cells (*Qiu et al., 2017*; *Trapnell et al., 2014*). We generated a subset of clusters using 'SubsetData' Seurat function using the raw counts. Combined progenitor clusters (F_earlyRG, F_midRG, F_lateRG, F_tRG, F_IPC, F_OPC) were selected. All clusters known to be generated from a non-cortical origin including microglia, endothelial, and mural cells were removed prior to the pseudo-time analysis. The expression matrix and a metadata file that contained above cluster information defined by Seurat were used as input for monocle package (*Supplementary file 2*).

The 'differentialGeneTest' function was used to select genes for dimensionality reduction (fullModelFormulaStr = '~Subtype.combined'). Top 1000 genes were then applied for cell ordering. The visualization of the minimum spanning tree on cells was obtained by Monocle2 'plot_complex_cell_trajectory' function. To visualize the ordered cells on Seurat's UMAP plot, we extracted the branch information for cells and added as metadata of the merged Seurat object ('AddMetaData' Seurat function).

## 7) Processing external scRNA-seq datasets for comparisons and preprocessing

scRNA-seq data of developing human brain from *Nowakowski et al., 2017*, from *Bhaduri et al., 2021*, were used for cross-species comparisons (*Figures 6 and 7*, *Figure 6—figure supplement 1*, *Figure 7—figure supplement 1*). All cells included in the analysis and cluster assignments from the provided matrix were mapped to the cell-type assignments provided in the metadata of Nowakowski

et al. dataset and the cell-type labels were used as provided by the authors without modification (*Supplementary file 5A*). For Bhaduri et al. dataset, GW25 sample was used for clustering by Seurat package as described above. For both, we removed low-quality cells and clusters annotated based on anatomical regions other than somatosensory cortex, and removed cells with less than 200 features. For Bhaduri et al. dataset, we filtered to cells with at least 200 features, less than 6500 features and less than 5% mitochondrial genes. This resulted in 25,485cells from around 180,000cells in Bhaduri et al. dataset, and in 2673cells in Nowakowski et al. dataset.

For organoid datasets, *Herring et al., 2022*, and *Bhaduri et al., 2020*, were used (*Figure 7— figure supplement 2*) with default settings in Seurat. The data driven from organoids at 5months, 9months, 12months and organoids at 3weeks, 5weeks, 8weeks, 10weeks were used for Herring and Bhaduri datasets, respectively.

## 8) Integrated analysis of human and ferret scRNA-seq datasets
Human datasets published by *Bhaduri et al., 2021*, and *Nowakowski et al., 2017*, were used for integration analysis with our ferret dataset using Seurat CCA (*Stuart et al., 2019*). First, a Seurat object was created for all datasets as described previously. Briefly, each object was individually processed with a normalization and variable features were identified. For human datasets, we excluded cells or samples obtained from anatomical regions other than somatosensory cortex. Then, the integration anchors between human and ferret Seurat objects were identified using 'FindIntegrationAnchors' Seurat function with default settings. The integration was then performed with 'IntegrateData' Seurat function. Scaling, PCA, and UMAP dimensional reduction were performed to visualize the integration results.

## 9) Cluster correlation analysis between ferret and human clusters
Correlation analysis was performed as described by *Bhaduri et al., 2020*, with modifications. Briefly, lists of cluster marker genes obtained by 'FindAllMarkers' Seurat function on individual datasets were extracted prior to an integration (Supplementary file 5A), except for Nowakowski et al. dataset, for which the marker gene list was provided. A score based on the specificity and the enrichment of cluster marker genes was generated (defined as 'genescore'). A genescore value was obtained by the multiplication of the average fold change that represents the gene enrichment, with the percentage of the cells expressing the marker in the cluster (pct.1) divided by the percentage of the cells from other clusters expressing the marker (pct.2: specificity). This function was employed on the marker genes for a human subtype of interest in the space of marker genes for ferret NPC (RG, IPC, OPC, and EP) clusters. Genescores for both ferret and human were then obtained for all markers shared between human subtype of interest (i.e. oRG) and ferret NPC subtypes. We then applied cor.test function using Pearson method to estimate a correlation and the significance of the correlation for the obtained genescores between ferret and human samples. These resulting values were represented for NPC subtypes on heatmap plots.

## 10) Prediction of oRG-like/tRG-like cells by cross-dataset analysis
The cell-type marker genes for oRG and tRG cell types were extracted from each human scRNA-seq datasets. To predict human oRG-like and tRG-like cells in ferret sample more precisely, we removed mature neuron, microglia, and endothelia clusters, as well as clusters assigned as unknown, from individual datasets for an integration between ferret and human NPC clusters using Seurat package as described in the above section (*Stuart et al., 2019*). After identification of anchors between human and ferret Seurat objects using 'FindIntegrationAnchors', we extracted the information of each pair of anchors, including one ferret and one human cell. If the human cell in the pair belongs to oRG (tRG) cluster in human dataset, we consider the other ferret cell in the same pair as oRG (tRG)-like cell.

## 11) Cluster score calculation
Based on a genescore calculated for marker genes as previously described, a cell-type predictive model was generated. For both datasets, our custom-made R scripts were applied to generate matrices using either human oRG markers or tRG markers using our integrated subsets. In the space of matrices reduced with corresponding human marker genes, a score we defined as 'oRG score' or

'tRG score' was created by multiplying human genescore for each marker genes with the expression values of these genes in all cells present in the integrated subset data. The results were represented by beeswarm and boxplot.

## Evaluation of immunolabeled cell number and statistical analysis

The counting frames on images of immunolabeled sections consisted of 150µm of width regions of interests (ROIs). The numbers of immunolabeled cells within selected ROIs were manually counted using the 'cell counter' tool of FiJi software (*Schindelin et al., 2012*). The VZ was identified by its cell density visualized by DAPI, and according to its thickness measured by a vertical length from the ventricular surface at indicated stages (100µm at P5, 70µm at P10). The proportions of immunolabeled cells positive for various markers were calculated using the summed data counted within all ROIs. The data obtained from lateral cortices of somatosensory area was averaged from two to three sections.

For CRYAB-positive cells, we only counted cells with nuclei which were visualized by DAPI staining. The numbers of positive cells were counted using two or three different animals.

Quantification was followed by Wilcoxon test to assess statistical significance and data on each section were considered as n=1.

## Quantification of RNAscope in situ hybridization and immunostaining image

Images of RNAscope in situ hybridization and FOXJ1 immunostaining were captured by using an IX83 inverted microscope (Olympus) equipped with Dragonfly 200 confocal unit (Andor), Zyla Z4.2 sCMOS camera (Andor), and UPLXAPO 60XO objective lens (Olympus). Imaging was performed to satisfy Nyquist sampling (xy = 48nm/pixel by using 2x zoom optic and z=136nm interval) and 3D deconvolution was performed by ClearView-GPU option of Fusion software (Andor). Default deconvolution parameter is used except for number of irritation (n=24) and mounting media reflection index (RI = 1.46). For quantification of images, CellProfiler version 4.2.4 (*Stirling et al., 2021*) was used. To detect mRNA dots of RNAscope in situ hybridization images, a single Z section of deconvolved images were processed by EnhancingSpeckles module (10 pixels size) and then mRNA dots were detected by IdentifyPrimaryObjects module, adaptive Otsu two-class method, using default parameter except for size of adaptive window (size = 10). Dot's diameter less than 4 pixels were removed as a background noise. Cytoplasmic ROIs were generated from smoothed RNAscope in situ hybridization images by MedianFilter module (window size = 10) and manually drawn seed ROIs of DAPI staining images by using IdentifySecondaryObjects module (propagation adaptive Otsu two-class method, using default parameter). FoxJ1 immunostaining raw images were measured by MeasureImageIntensity module. Quantified data were visualized and clustered by using RStudio (version 2022.07.1+554) with R (version 4.0.3). Each of mRNA count data and protein staining intensity data was log transformed and normalized, and hierarchical clustering was done by using a Euclidean distance and complete clustering method and cut tree to four clusters. Each cluster expression was plotted and classified to single or double positive, or double negative of SPARCL1 and FOXJ1 populations.

## Acknowledgements

We thank Kazuaki Yamaguchi, Rohab F Abdelhamid, and Chiharu Tanegashima for sequencing RNA and genomic DNA and all members of the Laboratory of Cell Asymmetry for providing technical support and helpful discussions. We also thank an anonymous reviewer 1 of Review Commons for his/her constructive suggestions, and Fumiya Kusumoto for his continuous encouragement. This work was supported by JSPS KAKENHI (Grant Numbers 18H04003, 17H05779, 19H04791) and RIKEN. RIKEN funds FM. MB was a RIKEN International Program Associate. QW was supported by a JSPS Postdoctoral Fellowship and a RIKEN Special Postdoctoral Researcher Program.

## Additional information

### Competing interests

Shigehiro Kuraku: Reviewing editor, eLife. The other authors declare that no competing interests exist.

## Funding

| Funder | Grant reference number | Author |
|---|---|---|
| Japan Society for the Promotion of Science | 18H04003 | Quan Wu |
| Japan Society for the Promotion of Science | 17H05779 | Fumio Matsuzaki |
| Japan Society for the Promotion of Science | 19H04791 | Fumio Matsuzaki |
| RIKEN Center for Biosystems Dynamics Research | Research Fund | Fumio Matsuzaki |
| RIKEN | International Program Associate | Merve Bilgic |
| Japan Society for the Promotion of Science | postdoctoral Fellowship | Quan Wu |

The funders had no role in study design, data collection and interpretation, or the decision to submit the work for publication.

## Author contributions

Merve Bilgic, Conceptualization, Data curation, Formal analysis, Investigation, Methodology, Writing – original draft, Writing – review and editing; Quan Wu, Conceptualization, Data curation, Formal analysis, Investigation, Methodology, Writing – original draft; Taeko Suetsugu, Investigation, Methodology; Atsunori Shitamukai, Data curation, Methodology; Yuji Tsunekawa, Tomomi Shimogori, Mitsutaka Kadota, Shigehiro Kuraku, Hiroshi Kiyonari, Methodology; Osamu Nishimura, Validation, Methodology; Fumio Matsuzaki, Conceptualization, Writing – original draft, Writing – review and editing, Supervision

## Author ORCIDs

Merve Bilgic ⓘ https://orcid.org/0000-0001-7975-7062
Quan Wu ⓘ http://orcid.org/0000-0001-9622-5367
Osamu Nishimura ⓘ https://orcid.org/0000-0003-1969-2580
Shigehiro Kuraku ⓘ https://orcid.org/0000-0003-1464-8388
Fumio Matsuzaki ⓘ https://orcid.org/0000-0001-7902-4520

## Ethics

All procedures during animal experiments were performed in accordance with the legal and institutional ethical regulations of RIKEN Center for Biosystems Dynamics Research.

Reviewer #1 (Public Review): https://doi.org/10.7554/eLife.91406.3.sa1
Reviewer #2 (Public Review): https://doi.org/10.7554/eLife.91406.3.sa2
Author Response https://doi.org/10.7554/eLife.91406.3.sa3

# Additional files

## Supplementary files

• Supplementary file 1. Quality control, metadata, and cluster information, related to *Figure 1* and *Figure 2*. (A) Summary of statistics for each sequencing library aligned to our custom genome. (B) Summary of statistics for representative libraries aligned to MusPutFur 1.0 (NCBI) and our custom genome. (C) Metadata for each cell. (D) Cluster markers identified using 'FindAllMarkers' Seurat function and filtered by min.pct=0.25 and logfc.threshold=0.25. (E) Ratios of clusters in each sample.

• Supplementary file 2. Pseudo-time trajectory analysis related to *Figure 4*. (A) Pseudo-time values for all cells treated in monocle v2. (B) Differentially expressed genes between cells at different pseudo-time branches, obtained by 'FindMarkers' Seurat function.

• Supplementary file 3. Expression of *SPARCL1* and *FOXJ1* in ferret truncated radial glia (tRG) cells, related to *Figure 5*. Cells are sorted by decreasing expression level of *SPARCL1* or *FOXJ1* in 'sorting

by Sparcl1' or 'sorting by Foxj1' sheets, respectively. Cells are colored and sorted by their pseudo-time state in *Figure 4* in 'state composition Foxj1& cryab'.

• Supplementary file 4. Distance of P10 cells with *SPARCL1* mRNAs±, FOXJ1 protein± from the apical surface, related to *Figure 5*.

• Supplementary file 5. Human-ferret integration, metadata, cluster markers, related to *Figure 6*. (A) Metadata for each cell in the integrated dataset. Ferret data was integrated with *Nowakowski et al., 2017* (Science) dataset. (B) Cluster markers used for calculating genescore in *Figure 6C–D*, obtained from *Nowakowski et al., 2017*. (C) Correlation between ferret and human cluster markers as shown in *Figure 6C–D*. (D) Markers shared between ferret and human truncated radial glia (tRG). (E) tRG score for ferret and human cells as shown in *Figure 6F*. Markers in D were used for calculating tRG score in *Figure 3F*.

• Supplementary file 6. Cluster markers in the integrated dataset, related to *Figure 7*. (A) Markers identified using the 'FindAllMarkers' Seurat function. (B) Differentially expressed genes identified using the 'FindMarkers' Seurat function, between cluster 28 and cluster 7 of the integrated data. (C) Differentially expressed genes identified using the 'FindMarkers' Seurat function, between cluster 21 and cluster 7 of the integrated data. (D) Markers of human outer radial glia (oRG). (E) Markers of ferret oRG.

• Supplementary file 7. Extraction of total RNA from different tissues, used for the gene model reconstruction, related to *Figure 1*.

• MDAR checklist

## Data availability

Genome assembly and chromium linked- read sequences were deposited in the DDBJ under accession numbers BLXN01000001–BLXN01022349 and DRA010274. These data have been mirrored on NCBI (https://www.ncbi.nlm.nih.gov/sra/?term=DRA010274; https://www.ncbi.nlm.nih.gov/nuccore/2619203981, and https://www.ncbi.nlm.nih.gov/bioproject/PRJDB9960). The gene models are available from Figshare under the DOI: 10.6084/m9.figshare.12807032. Single-cell RNA-seq data,deposited in the DDBJ under the DRR Run accession number DRX480639~DRX48064 (https://ddbj.nig.ac.jp/public/ddbj_database/dra/fastq/DRA016/DRA016867/), with codes used for analysis in https://github.com/wuquan723/Ferret-single-cell-data-from-Matsuzaki-lab (copy archived at *Wu, 2023*). Any information required to obtain and reanalyze the data reported in this paper is available from the lead contact upon request.

The following datasets were generated:

| Author(s) | Year | Dataset title | Dataset URL | Database and Identifier |
|---|---|---|---|---|
| Wu Q | 2023 | Ferret single cell transcriptomics analysis | https://ddbj.nig.ac.jp/public/ddbj_database/dra/fastq/DRA016/DRA016867/ | DDBJ BioSample, DRA016867 |
| Kuraku Lab in National Institute of Genetics | 2023 | MPF_Kobe_2.0: Genome assembly and gene-model of ferret (Mustela putorius furo) | https://doi.org/10.6084/m9.figshare.12807032 | Figshare, 10.6084/m9.figshare.12807032 |
| Bilgic et al | 2023 | Mustela putorius furo strain:Sable (domestic ferret) | https://www.ncbi.nlm.nih.gov/bioproject/PRJDB9960 | NCBI Bioproject, PRJDB9960 |
| Nishimura O, Kuraku S | 2023 | Mustela putorius furo isolate YT2017F09, whole genome shotgun sequencing project | https://www.ncbi.nlm.nih.gov/nuccore/2619203981 | NCBI Nucleotide, BLXN00000000.1 |
| RIKEN_BDR | 2023 | HiSeq X Ten paired end sequencing of SAMD00229826 | https://www.ncbi.nlm.nih.gov/sra/?term=DRA010274 | NCBI Sequence Read Archive, DRA010274 |

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
