## [Editor Report · eLife assessment]

This is an **important** study that improves gene models for the ferret genome and identifies neural progenitors that are comparable to those found in developing human brains. The data are **convincing** and clearly presented. Of particular interest to the field, the work identifies enriched expression of FOXJ1 in late truncated radial glia, strongly indicating that towards the end of neurogenesis, these cells likely give rise to ependymal cells. The work is of interest to anyone studying the development of the nervous system, especially colleagues studying the evolution of development.

---

## [Referee Report · Reviewer #1 (Public Review)]

In this manuscript, Bilgic et al aim to identify the progenitor types (and their specific progeny) that underlie the expanded nature of gyrencephalic brains. To do this, they take a comparative scRNAseq (single cell transcriptomics) approach between neurodevelopment of the gyrencephalic ferret, and previously published primary human brain and organoid data.

They first improve gene annotations of the ferret genome and then collect a time series of scRNAseq data of 6 stages of the developing ferret brain spanning both embryonic and post-natal development. Among the various cell types they identify are a small proportion of truncated radial glial cells (tRGs), a population known to be enriched in humans and macaques that emerges late in neurogenesis as the RGC scaffold splits into an oRGC that contact the pial surface and a tRG that contacts the ventricular surface. They find that the tRGs consist of three distinct subpopulations two of which are committed to ependymal and astroglial fates.

By integrating these data with publicly available data of developing human brains and human brain organoids they make some important observations. Human and ferret tRGs have very similar transcriptional states, suggesting that the human tRGs too give rise to ependymal and astroglial fates. They also find that the current culture conditions of human brain organoids seem to lack tRGs, something that will need to be addressed if they are to be used to study tRGs. While the primary human data set did contain tRGs, the stage or the region sampled were likely not appropriate, and therefore, the number of cells they could retrieve was low.

The authors have spent considerable efforts in improving gene modeling of the ferret genome, which will be important for the field. They've generated valuable time series data for the developing ferret brain, and have proposed the lineal progeny for the tRGs in the human brain. Whether tRGs actually do give rise to the ependymal and astrogial fates needs to be validated in future studies.

---

## [Referee Report · Reviewer #2 (Public Review)]

Bilgic et al first explored cellular diversity in the developing cerebral cortex of ferret, honing in on progenitor cell diversity by employing FACS sorting of HES5-positive cells. They have generated a novel single cell transcriptomic dataset capturing the diversity of cells in the developing ferret cerebral cortex, including diverse radial glial and excitatory neuron populations. Unexpectedly, this analysis revealed the presence of CRYAB-positive truncated radial glia previously described only in humans. Using bioinformatic analyses, the investigators proposed that truncated radial glia produce ependymal cells, astrocytes, and to a lesser degree, neurons. Of particular interest to the field, they identify enriched expression of FOXJ1 in late truncated radial glia strongly indicating that towards the end of neurogenesis, these cells likely give rise to ependymal cells. This study represents a major advancement in the field of cortical development and a valuable dataset for future studies of ferret cortical development.

---

## [Author Response]

The following is the authors’ response to the original reviews.

**Reviewer #1 (Recommendations For The Authors):**
The improvement of the gene annotations of the ferret genome was an important part of this study, and so I would recommend that the authors have a results section and figure dedicated to documenting this.

Thank you so much for appreciating our efforts on improving gene models, which was indeed a critical part in this study. According to the reviewer’s suggestion, we added a new section to the main text, “Improvement of the gene model for scRNA-seq of ferrets” with a figure (Fig.1 C, D, E).

Are the references to figure S8A, B alright (line 306)? In fact, that entire figure was not well described or out of place. In general, unlike the rest of the manuscript, the section dealing with the human-ferret comparison was a little bit confusing, and the figure legends were not extremely helpful.Could the authors please revisit the main text and figure legends of this section for clarity?

We agree with the reviewer’s recommendation. We removed references to Figure S8A, B. In place of that, we explained the reason more carefully; “We chose a recently published human dataset (Bhaduri et al, 2021) for comparison, because this study containing GW25 dataset which included more tRG cells than previous studies that did not contain GW25 data. Furthermore, we used only data at GW25”

We also revised several parts in this section to understand more easily by additional explanations as well as in the legends of Fig. 7 and Fig. S8.

**Reviewer #2 (Recommendations For The Authors):**
I have a few very minor comments on the manuscript.I would caution the authors against claiming that they have demonstrated bona fide generation of ependymal cells from tRG cells. While the expression of FOXJ1 is a very good indication, they have not demonstrated the morphological transformation of a tRG cell into an ependymal cell.

We agree the reviewer’s opinion. We have never thought that we proved that tRG differentiates ependymal cells, but we consider that this is highly likely the case (We use the term “suggest” in the abstract). To prove this genetically, we extensively tried to knock the EGFP gene into the CRYAB gene by the CRISPR/Cas9 method, to be able to show the lineage relationship between tRG and ependymal cells. However, we have so far failed to do this for a year trial. We also tried to just label tRG with EGFP and follow it in the slice culture.

However, we failed to keep the slice in the culture until we observed the transition from tRG shape to the ependymal shape. It seems to be a slow process. What we could do was to observe the transition from single cilia to multi-cilia, which is part of the morphological transition from epithelial neural stem cells such as Radial Glia to an ependymal-like sheet form. To prove this transition from tRG to ependymal cells (and also astrocytes) is one of the most important issue which needs some new idea, technique or strategy.

There are several typos throughout the manuscript that I would recommend fixing for example, page5 line 123 says "OLIGO2" instead of "OLIG2"

Thank you so much. We carefully read and corrected typos. We wish we corrected all of them.

Besides these two points, the manuscript is already prepared to a high standard.

I really appreciate reviewersʼ efforts to finish reviews in a short time, responding to our request related to the first authorʼs thesis application.